# Ternary complex of Kif2A-bound tandem tubulin heterodimers represents a kinesin-13-mediated microtubule depolymerization reaction intermediate

Daria Trofimova[1], Mohammadjavad Paydar[2], Anthony Zara[1], Lama Talje[2], Benjamin H. Kwok[2] & John S. Allingham[1]

Kinesin-13 proteins are major microtubule (MT) regulatory factors that catalyze removal of tubulin subunits from MT ends. The class-specific "neck" and loop 2 regions of these motors are required for MT depolymerization, but their contributing roles are still unresolved because their interactions with MT ends have not been observed directly. Here we report the crystal structure of a catalytically active kinesin-13 monomer (Kif2A) in complex with two bent αβ-tubulin heterodimers in a head-to-tail array, providing a view of these interactions. The neck of Kif2A binds to one tubulin dimer and the motor core to the other, guiding insertion of the KVD motif of loop 2 in between them. AMPPNP-bound Kif2A can form stable complexes with tubulin in solution and trigger MT depolymerization. We also demonstrate the importance of the neck in modulating ATP turnover and catalytic depolymerization of MTs. These results provide mechanistic insights into the catalytic cycles of kinesin-13.

[1] Department of Biomedical and Molecular Sciences, Queen's University, Kingston, ON K7L 3N6, Canada. [2] Institute for Research in Immunology and Cancer (IRIC), Département de médecine, Université de Montréal, P.O. Box 6128, Station Centre-Ville, Montréal, QC H3C 3J7, Canada. Correspondence and requests for materials should be addressed to B.H.K. (email: benjamin.kwok@umontreal.ca) or to J.S.A. (email: allinghj@queensu.ca)

Microtubules (MTs) are dynamic protein polymers that grow and shrink by addition and loss of αβ-tubulin subunits at their ends[1]. A wide variety of regulatory factors control MT polymerization dynamics to allow rapid spatial remodeling of the MT cytoskeleton during the cell cycle[1,2]. This activity is essential for mitotic spindle assembly and chromosome segregation[3], and enables directional transport of intracellular cargoes[4]. Kinesin-13 proteins are major MT-destabilizing factors in higher eukaryotes and a specialized class of motor proteins in the kinesin superfamily[3]. Rather than moving directionally along MTs, kinesin-13s catalyze tubulin disassembly at MT ends[5]. This activity is produced by a unique motor core found in the central region of the protein, and is the basis for kinesin-13s' common designation as Kin-M (middle) or Kin-I (internal) kinesins[5,6].

Since the first seminal characterization of a kinesin-13 protein from *Xenopus* (XKCM1) as a MT depolymerase in 1996[5], we have gained substantial knowledge on kinesin-13 isoforms and their roles in MT dynamics regulation. We know that the four kinesin-13 members in humans, Kif2A, Kif2B, Kif2C (also known as MCAK), and Kif24, function in a wide range of biological processes, such as spindle assembly, chromosome segregation, MT-kinetochore attachment, and cilia formation[3]. Most, if not all, of these essential functions are associated with the ability of kinesin-13s to depolymerize MTs and alter their polymerization dynamics. However, our understanding of how these enzymes catalyze the disassembly of MTs is still limited.

In reconstituted in vitro systems, kinesin-13 proteins can rapidly target MT ends either directly or through one-dimensional diffusion[7], and this targeting is enhanced by kinesin-13's positively charged neck[8]. Once at MT ends, these enzymes induce protofilament bending, as electron micrographs of depolymerizing MT ends show massive curled protofilament peels[5]. It has been shown that each kinesin-13 motor core binds a single tubulin protofilament[9], and forms contacts through the motor domain that stabilize intra-dimer curvature[10,11]. Interestingly, the presence of the neck restricts motor domain binding to alternate tubulin dimers of curved tubulin protofilament rings[12], but the molecular basis for this, and its implications on MT depolymerization, have not been determined.

The catalytic depolymerization of MTs by kinesin-13 is an ATP-dependent process[5,9,13,14]. In the landmark paper, Desai et al.[5] found that AMPPNP-bound XKCM1 was enriched at MT ends, but also formed a high affinity stable complex with tubulin dimers; resolvable by size-exclusion chromatography (SEC). This led to the hypothesis that although ATP is needed for targeting MT ends, its hydrolysis occurs later after depolymerization to release the enzyme from the dissociated tubulin dimers for additional rounds of catalysis[15]. This idea is supported by the observation that ATP hydrolysis-defective mutants of human Kif2C are still capable of depolymerizing taxol-stabilized MTs[14,15]. In contrast, an alternative model has also been proposed in which ATP hydrolysis occurs on the MT polymers prior to tubulin release[9,13]. This is based on a series of kinetics and microscopy-based experiments showing that MT ends strongly stimulate ATP turnover[9]. Therefore, it remains an open question whether the ATP hydrolysis step is needed for dissociating tubulin dimers from MT polymers. Resolving this question is crucial in understanding the molecular basis of kinesin-13-catalyzed MT depolymerization.

Mechanism aside, another important fundamental question is: what constitutes a functional depolymerase? Full-length kinesin-13 proteins are dimeric[16,17], and yet monomeric kinesin-13 constructs composed of the conserved motor domain and the N-terminally located neck are fully capable of depolymerizing MTs[18–22]. Accordingly, the neck plus the motor domain (denoted NM hereafter) is sometimes referred to as the minimal domain (i.e., kinesin-13-NM). Kinesin-13s from lower eukaryotes may be an exception of the neck requirement for MT depolymerization[23]. Biochemical and mutagenesis studies of these functional units have shown that the key elements required for MT depolymerization are the conserved KVD motif within loop L2 of the motor core, which also contains the ATPase domain, and the positively charged kinesin-13 neck[11,21,22,24]. Although crystallization studies have given us a glimpse of the structure of the KVD motif of the motor domain[15,21,25], direct experimental data on how it may promote disassembly of adjacent tubulin dimers are lacking in these early publications. Moreover, the complete neck domain is either not present or not resolved in these structures.

To better understand the molecular mechanism by which these key structural elements of kinesin-13 proteins induce MT depolymerization, we present the X-ray crystal structure of a MT depolymerization competent kinesin-13 construct (Kif2A-NM) bound to a head-to-tail array of two tubulin dimers. We also define the role of ATP hydrolysis by kinesin-13 in the catalyzed depolymerization reaction. Our Kif2A-NM-tubulin complex structure reveals that an AMPPNP-bound Kif2A monomer can simultaneously bind to two tubulin dimers, and that this interaction is accompanied by severe bending of the longitudinally associated tubulin dimers, more so than any other αβ-tubulin structures reported to date. Our biochemical analysis suggests that this outward bending of tubulins, resembling the kinesin-13-catalyzed structural changes at MT ends, is a prerequisite for triggering depolymerization.

## Results

**Structure of the Kif2A-NM−tubulin tetramer complex.** Human Kif2A is a 706-amino-acid kinesin-13 that functions as a homodimer in vivo[26]. Its motor domain is situated between residues 223 and 553, and is flanked by long N- and C-terminal regions that mediate subcellular targeting and dimerization (Fig. 1a)[18,27,28]. A monomeric motor domain construct that includes the kinesin-13-specific neck (Kif2A-NM; amino acids 153–553) exhibits the ability to catalytically depolymerize MTs in vitro (Fig. 1b), similar to that of the full-length dimeric kinesin-13 as previously reported[7,13,18,28,29]. However, a motor domain construct lacking the majority of the neck (Kif2A-MD; amino acids 203–554) is incapable of depolymerizing MTs in vitro, consistent with published literature[21,22,29,30]. A prediction is that the neck of kinesin-13 motors acts as an additional tether to the MT wall[31], but its precise binding site and contribution to MT depolymerization proficiency are unknown.

To gain insight into the mechanism of MT depolymerization by kinesin-13, we set out to solve the crystal structure of a depolymerization competent Kif2A construct (Kif2A-NM) in complex with tubulin. In the presence of the non-hydrolyzable ATP analog AMPPNP, a mixture of Kif2A-NM, tubulin, and the tubulin-capping protein DARPin[32,33] (molar ratio of about 0.8:1:1) gave two species that eluted earlier than the tubulin: DARPin complex by SEC (Fig. 1c). Their apparent molecular masses were estimated to be 270 and 150 kDa based on molecular weight standards and both contained Kif2A-NM, tubulin, and DARPin according to SDS-polyacrylamied gel electrophoresis (SDS-PAGE) analysis (Fig. 1d). When we pooled and concentrated the 150 kDa peak fractions for crystallization, we observed that much of the smaller species converted into the larger species upon re-running SEC, indicating that the two are interchangeable (Supplementary Figure 1). Of the two peaks, higher quality

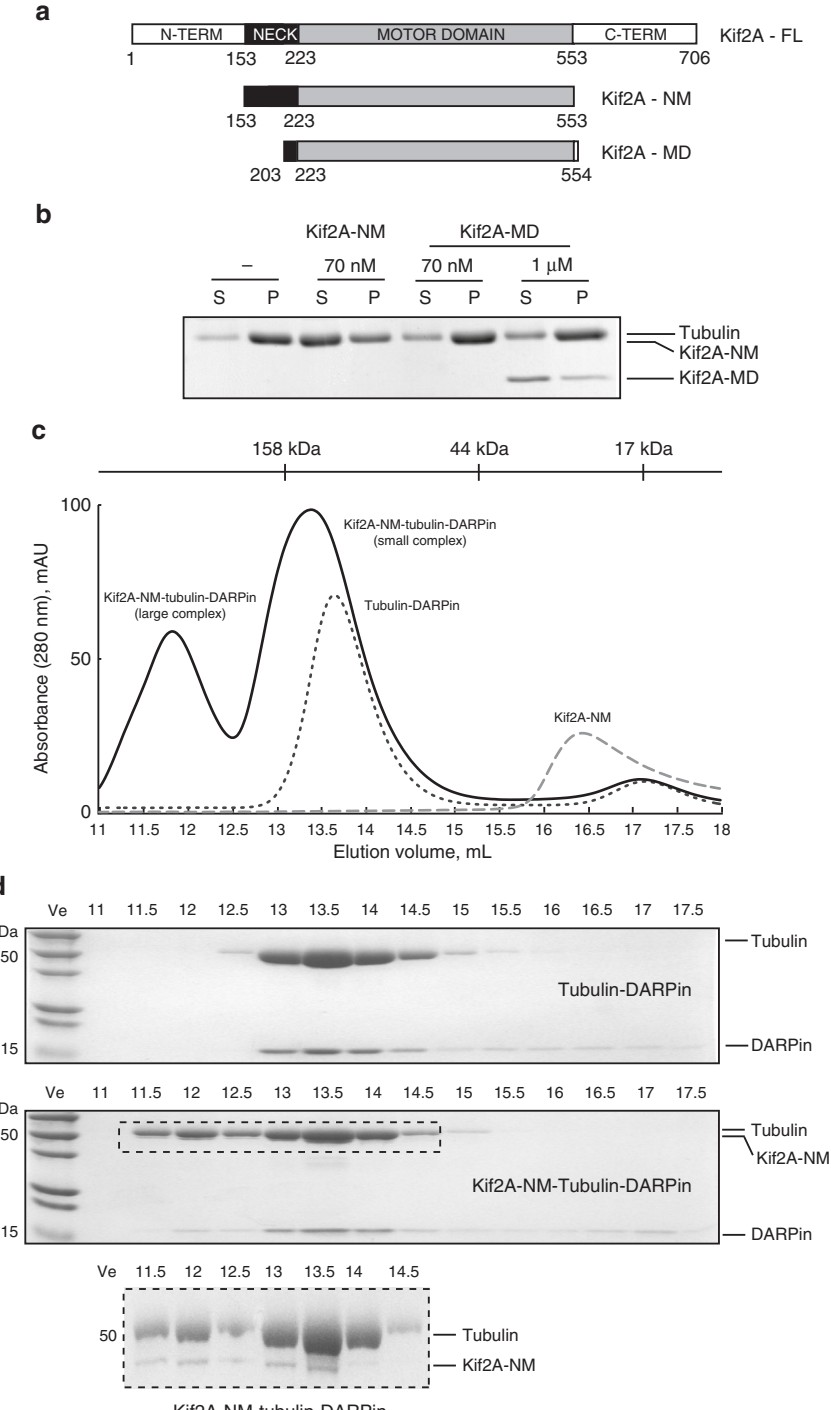

**Fig. 1** Functional analysis of Kif2A constructs. **a** Domain composition of human Kif2A and schematic of the Kif2A constructs used in this study. The length of the bar is proportional to the number of amino acids. **b** MT depolymerization activity of Kif2A constructs. Taxol-stabilized MTs were incubated with the indicated concentrations of Kif2A-NM and Kif2A-MD, or no kinesin for 10 min in BRB80 buffer. Free tubulin and MT polymers were separated into supernatant (S) and pellet (P) fractions by ultra-centrifugation-based sedimentation assay. Fractions were resuspended and boiled in Laemmli buffer. Equal portions were loaded and analyzed on a Coomassie blue-stained SDS-PAGE gel. **c** SEC profiles of Kif2A-NM alone (gray dashed line), tubulin–DARPin complex (black dotted line), and Kif2A-NM–tubulin–DARPin complex in 0.8:1:1 molar ratio (black solid line). All samples were supplemented with 0.1 mM AMPPNP and applied to an Superdex 200 10/300 GL column in HEPES buffer. **d** Twelve percent SDS-PAGE gels of elution fractions from the above experiments. The molecular weight of Kif2A-NM = 48 kDa, tubulin = 50 kDa, and DARPin = 18 kDa. The inset shows a 10% SDS-PAGE gel of fractions containing the Kif2A-NM–tubulin–DARPin complex in order to visualize the Kif2A and tubulin proteins. All uncropped gels are shown in Supplementary Figures 9, 10, 11, 12, 13 and 14

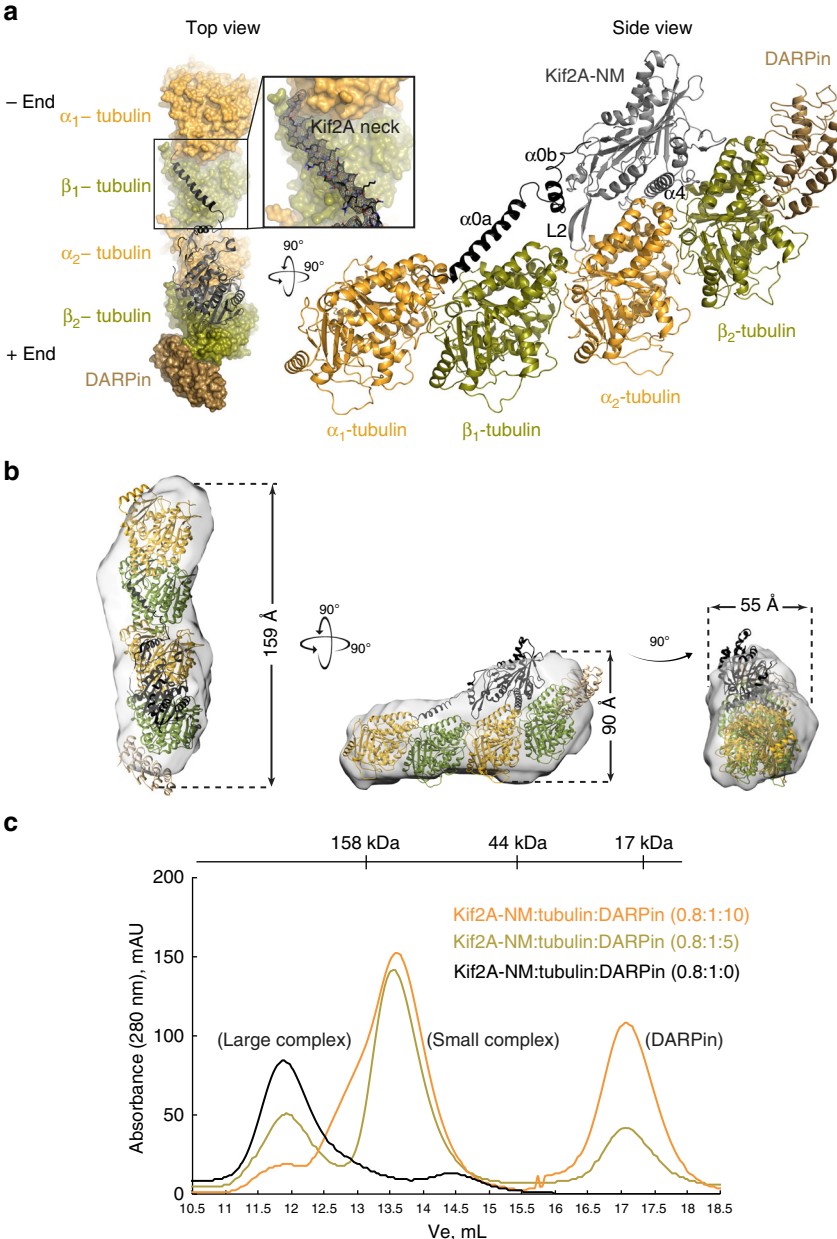

**Fig. 2** Structure of Kif2A-NM in complex with tubulin and DARPin. **a** The X-ray crystal structure of a Kif2A-NM–tubulin–DARPin complex is shown in two views (from the top and side of the tubulin filament). The Kif2A is in black; α-tubulin is in orange; β-tubulin is in green; and DARPin is in light brown. The insert represents the $F_{obs} - F_{calc}$ omit map (contoured at 3.0$\sigma$) of the neck of Kif2A-NM calculated after deletion of the neck from final model. All figures of structural models were generated with PyMOL[67]. **b** SAXS envelopes for the Kif2A-NM–tubulin–DARPin complex. The crystal structure of the complex shown in ribbons representation and superimposed onto envelopes (in gray) by Chimera[68]. **c** SEC profiles of mixtures of Kif2A-NM, tubulin, and DARPin at the indicated molar ratios. All mixtures were supplemented with 0.1 mM AMPPNP and passed through a Superdex 200 10/300 GL column in HEPES buffer

crystals grew from the 150 kDa one and these were harvested for X-ray diffraction.

The structure of the crystallized complex was determined by molecular replacement (MR) using the coordinates of the ADP-bound Kif2A motor domain (PDB ID: 2GRY), and the α-tubulin, β-tubulin, and DARPin subunits from the kinesin-1-tubulin complex (PDB ID: 4LNU[33]) as separate rigid bodies. The MR solution comprised one Kif2A-NM monomer bound to the concave side of a curved assembly of two αβ-tubulin dimers arranged head to tail and capped at the plus end by DARPin (Fig. 2a). The final structure was refined to a resolution of ~3.5 Å (Table 1), giving electron density maps that show well-connected main chain density for all subunits of the Kif2A-

NM–tubulin–DARPin complex. Notably, the N-terminal section (residues 157–195) of the neck of Kif2A can be seen interacting with one of the tubulin dimers (labeled α₁β₁; buried surface area = 915 Å²). The other dimer (labeled α₂β₂) is occupied by the Kif2A motor domain, which contains MgAMPPNP in the active site. As predicted from molecular modeling studies[15,21], the tip of the β-hairpin of loop 2 of the motor domain reaches into the inter-dimer interface to interact with both α₂ and β₁ tubulin.

To confirm that the subunit arrangement in the crystallographic complex is also formed in solution, we performed in-line SEC and small-angle X-ray scattering (SEC-SAXS) experiments on the contents of the 270 kDa peak. The SAXS data show that the shape of the molecular envelope has nearly identical dimensions to the

**Table 1 Data collection and refinement statistics**

|  | Kif2A–tubulin–DARPin |
|---|---|
| **Data collection**[a] |  |
| Space group | P2$_1$ |
| Cell dimensions |  |
| $a, b, c$ (Å)[b] | 145.05, 82.19, 147.66 |
| $\alpha, \beta, \gamma$ (°) | 90.0, 109.33, 90.0 |
| Resolution (Å) | 49.22–3.51 (3.64–3.51) |
| Unique reflections | 39,552 (2673) |
| $R_{meas}$ | 0.249 (2.685) |
| $I/\sigma I$ | 7.28 (0.47) |
| Multiplicity | 3.6 (3.1) |
| Completeness (%) | 95.61 (65.61) |
| **Refinement** |  |
| Resolution (Å) | 49.22–3.51 |
| No. reflections | 39,550 (2725) |
| $R_{work}/R_{free}$ | 0.236/0.287 |
| No. of atoms |  |
| Protein | 17,634 |
| Ligand | 160 |
| Water | 0 |
| B factors |  |
| Protein | 146.35 |
| Ligand | 135.16 |
| Coordinate error (Å) | 0.7 |
| R.m.s.d. |  |
| Bond lengths (Å) | 0.0039 |
| Bond angles (°) | 0.71 |
| Ramachandran |  |
| Favored region (%) | 88.56 |
| Allowed region (%) | 9.84 |
| Outliers (%) | 1.60 |

[a]Data were collected on a single crystal
[b]Values in parentheses are for the highest-resolution shell

crystallized 1:2:1 Kif2A-NM–tubulin–DARPin complex (Fig. 2b). It is elongated with a maximal dimension ($D_{max}$) of 195 Å, and is asymmetric, with a bulge on one end where Kif2A is located. The radius of gyration (Rg) was estimated to be 55.9 Å, and the modeling fit was validated by comparing the experimental and theoretical SAXS profiles. The theoretical scattering curve obtained for the crystallized complex fits the experimental SAXS profile with a $\chi^2$-value of 1.09 (Supplementary Figure 2 and Supplementary Table 1). This result demonstrates that Kif2A-NM can bind to two αβ-tubulin dimers in solution.

To determine the relationship between the molecular assemblies in the SEC peaks, we varied the molar ratio of DARPin relative to Kif2A-NM and tubulin in the assembly mixture. We found that addition of excess DARPin favored formation of the small complex over the large one (Fig. 2c). This implies that excess DARPin is able to wedge apart the tubulin dimers held together by Kif2A-NM, thereby releasing the one associated with the neck domain. However, when DARPin is in lower abundance, such as in the experimental condition during SEC to isolate the ternary complex (Fig. 1c), the head-to-tail array of tubulin dimers can form a stable complex with AMPPNP-bound Kif2A-NM. These findings suggest that excess DARPin can compete Kif2A-NM off the tethered double tubulin dimers. This discovery would prove useful for understanding how neck domain interaction with tubulin contributes to catalytic activity of Kif2A-NM, as described below.

**Interactions of the Kif2A neck with tubulin**. Although the electron density map did not provide coverage of all the amino-acid side chains in the model, there was adequate density for many bulkier residues to permit unambiguous assignment of the helical register of the neck (Fig. 3a, b). The neck contacts both subunits in the α$_1$β$_1$-tubulin dimer (Fig. 3b). Residues 157–159 form a random coil that interacts with the C-terminal end of helix H11 in α$_1$-tubulin and the C terminus of β$_1$-tubulin. Interestingly, this surface of the tubulin dimer is not an interface for the motor domain of kinesins. Residues 160–185 form a well-defined helix (α0a) that runs antiparallel to helix H12 of β$_1$-tubulin. Many of the positively charged residues in α0a juxtapose the large patch of negatively charged residues on the surface of β$_1$-tubulin. This supports previous observations that point mutations that preserve the positive charge richness of the neck retain depolymerization function[22]. Additional neck-stabilizing interactions include the non-polar side chain of Val160 and several negatively charged (Glu164, Glu171, and Glu182) and polar residues (Asn158, Gln167, and Gln178) that contact the buried helix H12 of β$_1$-tubulin.

Helix α0a is brought into alignment with helix H12 of β$_1$-tubulin by a second, shorter helix (α0b; residues 195–208; Fig. 3a) that forms alongside the β-hairpin of loop 2 (Fig. 3b). Helices α0a and α0b are connected by a short, kinked loop (formed by residues 188–194) that changes the direction of the neck. They also interact via a salt bridge between Arg181 and Glu196. This conformation of α0b is almost identical to that of ADP-Kif2C (PDB ID: 2HEH) and the short loop 2 construct of Kif2C (PDB ID: 4Y05)[15,21], suggesting that α0a of Kif2C aligns along H12 of β-tubulin as well. The other reported conformations of α0b in the Kif2C–tubulin complex (PDB ID: 5MIO)[25] and the "activated conformation" of the mouse Kif2C core (PDB IDs: 5XJA and 5XJB)[34] indicate that the neck can dissociate from the distal tubulin-binding site whilst the motor core is engaged with tubulin (Supplementary Figure 3). The purpose, if any, of these conformations within the kinesin-13 catalyzed MT depolymerization cycle is uncertain given the absence of proper context (e.g., an adjacent tubulin dimer), and will likely require alternate experimental settings to resolve.

**Movement of the longitudinal tubulin interface**. The Lys262, Val263, Asp264 triplet (KVD motif) of Kif2A interacts with both β$_1$-tubulin and α$_2$-tubulin at the inter-dimer interface (Fig. 3c). Val263 inserts into a hydrophobic pocket formed by the non-polar atoms of Tyr262, Pro263, Arg264, Ile265, Asp431, Glu434, and Val435 of α$_2$-tubulin, while the C$_\alpha$ and C$_\beta$ atoms of Asp264 of Kif2A are nearest to Tyr262 of α$_2$-tubulin and Lys402 of β$_1$-tubulin. Adjacent to the KVD triplet, Leu265 interacts with His406 (β$_1$) and Tyr262 (α$_2$), and Thr266 is in position to H-bond with the ε-amino group of Lys402 (β$_1$). Although Lys262 and Asp264 are close to acidic and basic resides of α$_2$ and β$_1$, respectively, H-bonds or salt bridges are not observed in the Kif2A-NM–tubulin–DARPin complex in the manner predicted previously by computational modeling[15]. Rather, the side chain of Lys262 appears to extend away from the tubulin surface, and Asp264's side chain turns inward to form an intramolecular H-bond with Thr266. Interestingly, these interactions and the overall L2 loop conformation are remarkably similar to those of the Kif2C–tubulin complex (PDB ID: 5MIO)[25]. Given the recent observation that interchanging the lysine/glutamate residues of the Kif2C KVD motif compromised MT depolymerization activity[15], it is possible that the importance of these residues play out at an earlier stage of MT binding than has been captured in the Kif2A-NM–tubulin–DARPin complex.

Directly next to the KVD insertion site, there is noticeable movement of the T7–H8 motif of α$_2$-tubulin away from its position in the Kif2C–tubulin complex (Fig. 3d; the root mean squared deviation (r.m.s.d.) of the Cα positions of this motif is 3.0 Å and maximum Cα displacement is 6.1 Å). This finding is important because the T7–H8 motif forms a major longitudinal interface in protofilaments and is presumed to act as a cohesive

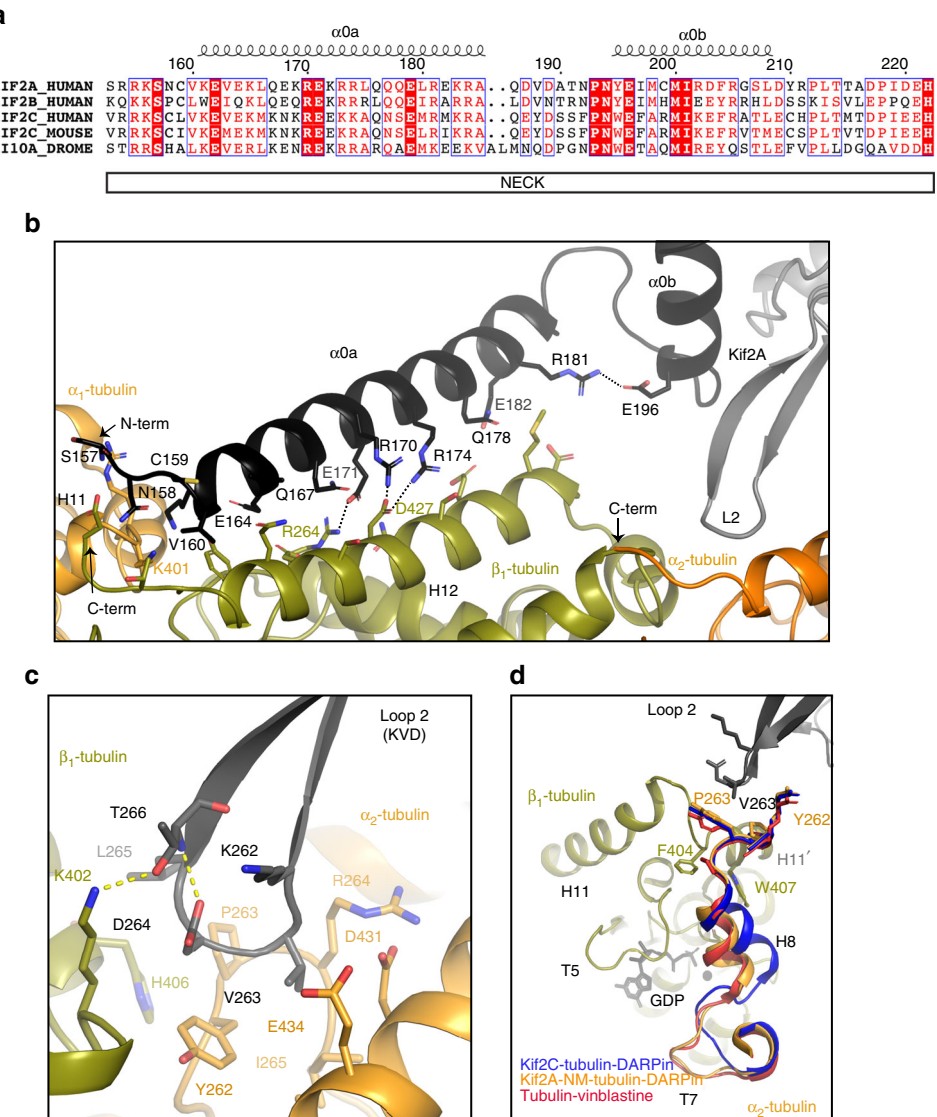

**Fig. 3** Kif2A neck and loop 2 interactions with tubulin. **a** Sequence alignment of selected kinesin-13 motors was performed by Clustal Omega[69], and the assignment of the secondary structure for the neck was performed using ESPript[70] according to the tubulin-bound Kif2A crystal structure. **b** Regions in $\alpha_1$-tubulin (orange) and β1-tubulin (green) that interact with the Kif2A neck helix (black) are shown in cartoon format, with selected side-chain residues shown as sticks. **c** Interactions of loop 2 of Kif2A with $\beta_1$ (green) and $\alpha_2$-tubulin (orange) at the inter-dimer interface. **d** Conformation of the T7–H8 motif of $\alpha_2$-tubulin in the Kif2A-NM–tubulin–DARPin complex (orange) is shown as a view looking through $\alpha_2$-tubulin, toward the nucleotide pocket of β1-tubulin (green). T7–H8 of the Kif2C–tubulin–DARPin complex (PDB ID: 5MIO; blue) and stathmin–tubulin–vinblastine complex (PDB ID: 4EB6; red) are shown, along with the position of KVD motif of Kif2A, after superposition using PyMol

structural unit with the T3 and T5 loops of the opposing β-tubulin subunit[35]. Curved tandem tubulin complexes formed by stathmin-like proteins do not exhibit this dramatic deformation either, unless they are complexed with MT-depolymerizing agents that lie close to the T7–H8 motif, such as vinblastine (Fig. 3d, the r.m.s.d. of Cα positions is 0.8 Å relative to PDB ID: 4EB6)[36–39]. It is therefore possible that positioning the tip of loop 2 near this motif helps elicit a T7–H8 conformational change as part of the protofilament bending mechanism by kinesin-13. This change was not observed in the Kif2C–tubulin complex[25], presumably due to the lack of an adjoining tubulin dimer.

**Conformation of the tubulin-bound Kif2A motor domain.** In the complex, Kif2A-NM is bound to the non-hydrolyzable ATP analog AMPPNP. Superimposing isolated (tubulin-free) ADP-Kif2A (PDB ID: 2GRY) via the P-loop shows a considerable

reconfiguration of Switch 1 and 2 (Fig. 4a), similar to AMPPNP-bound Kif2C–tubulin and many AMPPNP-bound motile kinesins[25,40–42]. The two universally conserved serine residues in Switch 1 (Ser431 and Ser433) are within distance to establish hydrogen bonds with the terminal nucleotide phosphate and bound $Mg^{2+}$, and loop L11 is now fully ordered in the Kif2A-NM–tubulin–DARPin complex. Loops L8 and L12 (and the β5a–β5b hairpin) also change position in the AMPPNP-Kif2A-NM motor core to interact with helices H8 and H12 of $\beta_2$-tubulin (Fig. 4b). The same is true for α4 and L11, which, along with α6, contact the H3, H11, and H12 helices of $\alpha_2$-tubulin. The most striking change from isolated Kif2A involves a ~26° rotation of the α4 and α5 helices, allowing L12 to engage helix H12 of $\beta_2$-tubulin. A nearly identical rotation of α4 and α5 was observed in the Kif2C–tubulin complex, adding support to the proposal that the tubulin-bound conformation of kinesin-13s is similar to that of kinesin-1 in its tubulin-bound state[15,25,33,40], as emphasized in

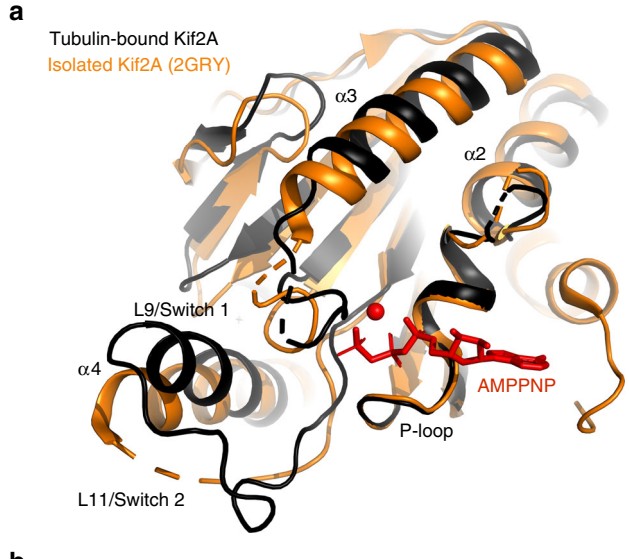

**a**

Tubulin-bound Kif2A
Isolated Kif2A (2GRY)

α3

α2

L9/Switch 1

α4

AMPPNP

P-loop

L11/Switch 2

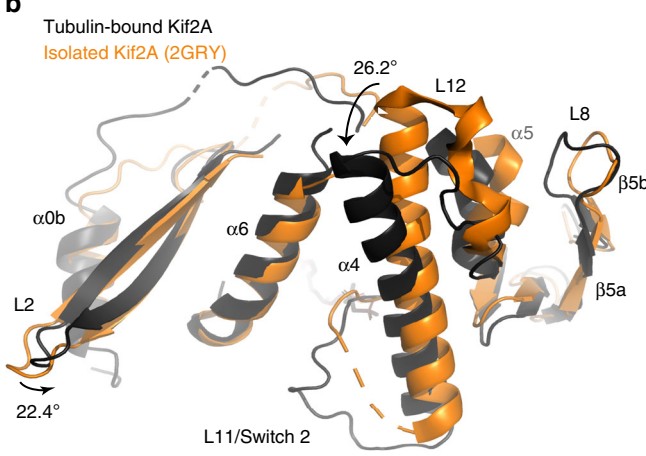

**b**

Tubulin-bound Kif2A
Isolated Kif2A (2GRY)

26.2°

L12

L8

α5

β5b

α0b

α6

α4

β5a

L2

22.4°

L11/Switch 2

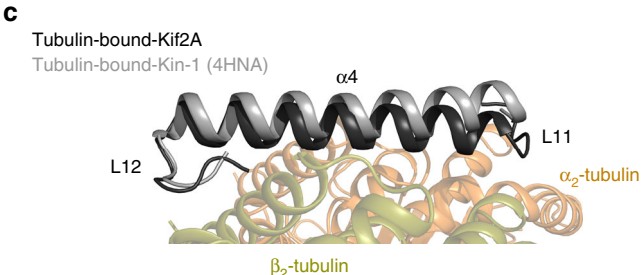

**c**

Tubulin-bound-Kif2A
Tubulin-bound-Kin-1 (4HNA)

α4

L12

L11

α2-tubulin

β2-tubulin

**Fig. 4** Conformation of the Kif2A motor domain. **a** View of the nucleotide-binding pocket of tubulin-bound Kif2A (black) superimposed on isolated ADP-Kif2A (orange; PDB ID: 2GRY) via the P-loop. **b** View of the tubulin-binding surface. **c** Comparison of the α4 helix orientation in the tubulin-bound Kif2A (black) and tubulin-bound Kin-1 (PDB ID: 4HNA, gray). Tubulin is represented in cartoon format

Fig. 4c. These findings argue against postulations that a more curved tubulin interface on the kinesin-13 motor domain explains their MT depolymerization capability[12,21]. Although the recent mouse Kif2C core:ADP-AlF$_x$ and Kif2C core:ADP-BeF$_x$ structures (PDB ID: 5XJA and 5XJB, respectively) were used to support this hypothesis and are presented as the "activated" conformation that sustains the Kif2-tubulin 1:2 complex[34], the tubulin-binding elements of these structures (L8, β5a–β5b β-sheet, L11, L12, α4, α5, and α6) superimpose poorly with those of Kif2A-NM (Supplementary Figure 4).

**Curvature of the Kif2A-NM-bound tubulin oligomer**. A notable feature in our Kif2A–tubulin ternary structure is that each tubulin subunit is related by a distinct bend angle (Fig. 5, side view), indicating that the interactions of the neck, L2 loop, and motor domain of Kif2A each affect tubulin polymer curvature in different ways. The curvature of α$_2$β$_2$-tubulin (12.1°) marginally exceeds the Apo and ADP–AlF$_4$-bound kinesin-1 complexes (11.6° and 9.2°, respectively)[33,40], and is less curved than the Kif2C-DARPin fusion (14.7°)[25]. In contrast, the tubulin dimer bound by Kif2A's neck is much more curved (15.3°), and there is an even larger bend at the inter-dimer interface (15.8°). This implies that the kinesin-13-specific neck and the extended L2 loop of Kif2A are the major contributors to longitudinal tubulin bending. Combined with the curvature at α$_2$β$_2$-tubulin, the tethered tubulin complex formed by Kif2A-NM is more curved than any other tubulin structure (see examples in Supplementary Figure 5).

The Kif2A-NM–tubulin–DARPin complex also shows a rotational displacement of β-tubulin relative to α-tubulin in both dimers (Fig. 5, top view). While this has been described for cryo-electron microscopy (cryo-EM) structures of tubulin rings formed by the *Drosophila* kinesin-13 KLP10A[11], and for stathmin–tubulin complexes containing vinca-domain ligands[36], the divergence from linearity is more pronounced for Kif2A-NM-bound tubulin (Fig. S5). This, in combination with the substantial outward curvature of tubulin subunits, would be incompatible with the straight lateral contacts within the MT lattice.

By superimposing each Kif2A-NM-bound tubulin subunit onto the corresponding subunits of the less-curved stathmin–tubulin–cholchicine complex (PDB ID: 1SA0), subtle structural differences emerge that could explain how additional curvature and rotation are accommodated in the Kif2A-NM–tubulin complex. One involves the shift in the T7–H8 helix of α$_2$-tubulin described earlier. Coincidental with this change are subtle adjustments of the side-chain dihedral angles of Phe404, His406, and Trp407 in H11′ of β$_2$-tubulin. These residues form a localized pocket of non-polar side chains of helix H8 at the inter-dimer interface. Similar changes are visible, albeit less pronounced, at the intra-dimer interface of α$_1$β$_1$-tubulin. In β$_1$-tubulin, there is also a slight upward shift of the helix H12 toward Kif2A, along with movement of the H10-S9 loop ~0.6 Å closer to α$_1$-tubulin. At the α$_2$β$_2$-tubulin, where curvature is the lowest, there is barely any change.

**ATP turnovers are modulated by neck binding to tubulin**. To evaluate the relevance of the 1:2 Kif2A–tubulin complex in terms of Kif2A's function as an enzyme, we measured the tubulin-stimulated ATPase activity of the Kif2A-NM and Kif2A-MD constructs in the presence or absence of excess DARPin. We postulated that if binding of the Kif2A neck to a second α$_1$β$_1$-tubulin subunit participates in the tubulin-stimulated turn-over of ATP, Kif2A-NM activity should be affected by the presence of excess DARPin because DARPin and the neck cannot bind to the α$_1$β$_1$-tubulin dimer simultaneously. Indeed, we observed that excess DARPin significantly suppressed tubulin-stimulated ATPase activity of Kif2A-NM (Fig. 6a). In contrast, the Kif2A-MD construct, which only forms a 1:1 complex with tubulin (Supplementary Figure 6), was not affected by DARPin. To verify that our observation is true for other kinesin-13s, we performed this ATPase experiment with human Kif2C and observed an identical result (Fig. 6b). This suggests that formation of the 1:2 kinesin–tubulin complex is also true for Kif2C/MCAK.

**MT-stimulated ATP turnover and tubulin disassembly**. Given the competing nature of DARPin and kinesin-13-NM in binding to tubulin dimers as demonstrated by the dimer-stimulated

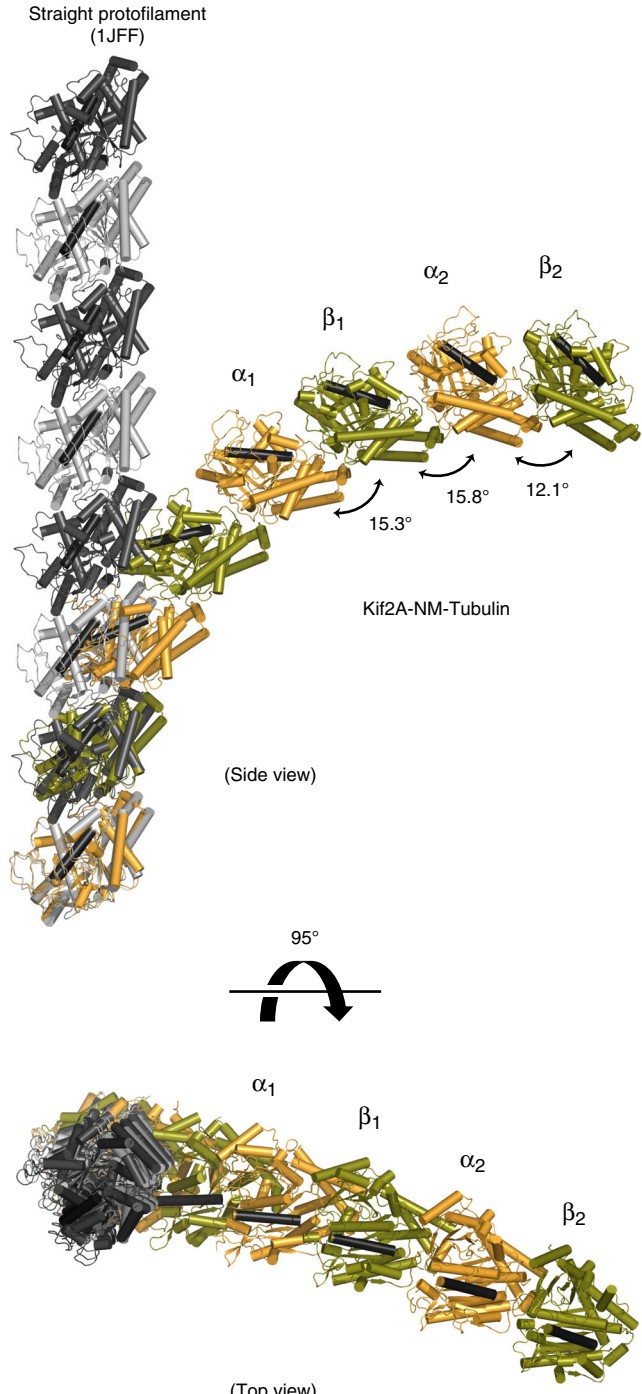

**Fig. 5** Curvature and rotational displacement of tubulin induced by Kif2A-NM. Views are from the side and looking down the long axis of the straight protofilament model. Tubulin subunits of the Kif2A-NM–tubulin–DARPin complex (Kif2A and DARPin are not shown) are colored orange (α-tubulin) and green (β-tubulin), within which a single helix is colored black as a reference point. Straight protofilament structure (α-tubulin in light gray, β-tubulin in dark gray PDB ID: 1JFF) is shown for reference. Extended protofilaments were generated by aligning additional copies of each tubulin complex in an overlapping fashion. The degree of rotation between α- and β-tubulin subunits in the Kif2A-NM–tubulin–DARPin complex was determined from the transformation required to superimpose each subunit within, or between, each heterodimer using the RotationAxis plugin in PyMOL (https://pymolwiki.org/index.php/RotationAxis)

ATPase activity (Fig. 6a, b), we wondered if we could use DARPin to probe for the catalytic cycles of kinesin-13-mediated MT depolymerization, in particular, the relationship between ATP hydrolysis of kinesin-13 and tubulin disassembly. Since DARPin's binding site is masked when tubulins are incorporated into the polymer (except those at the ends) we anticipated that it would have no or limited effect on MT-stimulated activity of kinesin-13. To verify this, we measured the average MT-stimulated ATPase rates of Kif2A-NM and Kif2A-MD in the absence or presence of DARPin with excess MTs (at [tubulin dimer] = 2 μM) over a 10-min reaction time. We chose the 10-min reaction time because of the relatively constant turnover rate over this time period when MTs were still in excess (i.e., when [tubulin] > 1 μM). To our surprise, we found that DARPin significantly suppressed MT-stimulated ATPase activity of Kif2A-NM, but not that of Kif2A-MD (Fig. 6c). Identical results were obtained for Kif2C-NM and Kif2C-MD (Fig. 6d). Because of this unexpected result, we considered the possibility that DARPin might potentiate, instead of inhibit, Kif2A-NM- and Kif2C-NM-induced MT depolymerization resulting in MT concentration falling below 1 μM (at which MTs became limiting in stimulating ATP turnover of Kif2A-NM, and would result in a decrease in ATPase rate). To address this, we ran time course experiments at 2.5-min intervals to monitor ATPase activity of Kif2A-NM over time. As we anticipated, the ATPase rates of our control experiment without DARPin stayed unchanged over the 10-min time period. In contrast, the initial ATPase rate of Kif2A-NM in the presence of DARPin (both 2 and 20 μM) did not deviate much from that of the control, but the rate diminished over time (Fig. 6e). To confirm that the time-dependent decrease in ATPase rate was due to the loss of polymers over time, we removed portions of the reactions in some of our experimental sets (n = 3) and subjected them to ultra-centrifugation to separate MTs and the dissociated tubulin dimers. The results from these sedimentation experiments indeed confirmed our hypothesis (Fig. 6f). We found that DARPin potentiated the effect of Kif2A-NM and that the excess loss of MTs could account for the decrease in ATPase rates in the presence of DARPin at later time points.

Our biochemical analyses of Kif2A-NM with DARPin presented an interesting scenario: DARPin interfered with Kif2A-NM binding to tubulin and inhibited its tubulin dimer-stimulated ATP turnover, but potentiated its effect on MT depolymerization. This brought back an important question: what is ATP hydrolysis actually needed for? Our Kif2A-NM–tubulin–DARPin structure suggests that Kif2A-NM binding induced curvature of adjacent tubulin dimers in the presence of AMPPNP that could be sufficient to trigger tubulin dissociation from MT polymers. However, published data from vertebrate kinesin-13s indicated that AMPPNP is insufficient to support catalytic depolymerization of MTs[15]. Therefore, ATP hydrolysis is likely needed either for depolymerization or for releasing kinesin-13s from the dissociated tubulin dimers to ensure catalytic removal of tubulin dimers from MT ends. Our SEC data suggested excess DARPin could dissociate the 1:2 Kif2A-NM:tubulin complex into 1:1 complex instead. Deducing from this result, we postulated that excess DARPin might mimic ATP turnover-mediated Kif2A-NM dissociation from the 1:2 complex with adjacent tubulin dimers from MT ends. To test this, we performed Kif2A-NM-induced MT depolymerization assays in the presence of DARPin with AMPPNP or ADP (as a control). Consistent with published literature[15,34], Kif2A-NM was not able to induce substantial MT depolymerization with either AMPPNP or ADP (Fig. 7a). Remarkably, in the presence of DARPin, Kif2A-NM was able to depolymerize MTs with AMPPNP, but not with ADP. Similar result was obtained using Kif2C-NM (Fig. 7b). Importantly, DARPin alone (from 0.25 to 2 μM) has negligible

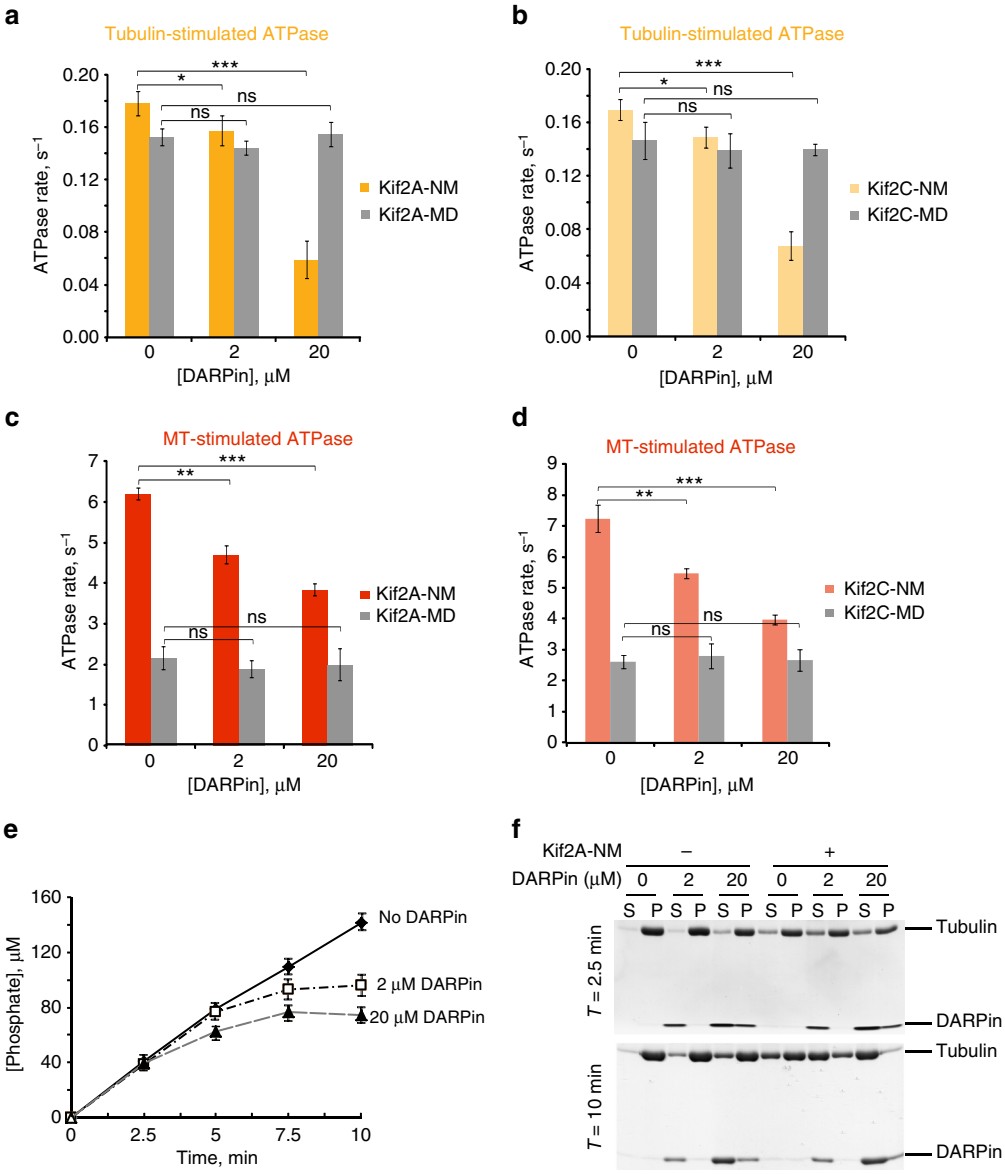

**Fig. 6** MT-stimulated ATPase and MT-depolymerizing activities of kinesin-13s. **a**, **b** Effect of DARPin on tubulin-stimulated ATPase activity of kinesin-13 proteins. ATP turnover rates of Kif2A-NM (orange) and Kif2A-MD (gray) in the presence of tubulin with the indicated concentrations of DARPin are shown in **a**. Equivalent data for Kif2C-NM (light orange) and Kif2C-MD (gray) are shown in **b**. **c**, **d** Effects of DARPin on MT-stimulated ATPase activities of kinesin-13 proteins: **c** ATP turnover rates by Kif2A-NM (red) and Kif2A-MD (gray) in the presence of 2 μM of taxol-stabilized MTs and the indicated concentrations of DARPin are shown. **d** Corresponding data sets of experiments for Kif2C-NM (light red) and Kif2C-MD (gray) as shown. Data from **a** and **b** represent averages from at least four independent experiments. Error bars, S.D.; ns: $p > 0.05$; *$p \leq 0.05$; **$p \leq 0.01$; ***$p \leq 0.001$, by Student's $t$-test. **e** Time course plots of inorganic phosphate release by Kif2A-NM under the same experimental conditions as in **c**. Averages from $N = 3$. Error bars, S.D. **f** Monitoring of MT polymers present at the 2.5 and 10-min time points of the experiment shown in **c** (right three pairs) and the corresponding controls without Kif2A-NM (left three pairs) by a centrifugation-based sedimentation assay. S supernatant fraction; P pellet fraction. Samples were resolved by SDS-PAGE and gels were stained with Coomassie blue. Representative gels are shown

effect on MT depolymerization (Supplementary Figure 7). Given that AMPPNP is a non-hydrolyzable analog of ATP, our result indicated that ATP binding, rather than its hydrolysis, is needed to induce MT depolymerization. It also suggests that ATP hydrolysis is required for releasing the bound Kif2A-NM from the dissociated tubulin dimers, and that the presence of DARPin bypasses this requirement by releasing the enzyme from the bound tubulin dimers, likely via competition with its neck binding to $\beta_1$-tubulin (refers to illustration in Fig. 2a). We reasoned that although DARPin does not bind to the same site as the Kif2A neck, its association with $\beta_1$-tubulin is incompatible with simultaneous binding of the neck helix and thereby

displacing the $\alpha_1\beta_1$-tubulin dimer from the ternary complex. This interpretation is consistent with our observation in the SEC experiments that only AMPPNP, and not ATP or ADP, is able to produce stable 1:2:1 Kif2A-NM–tubulin–DARPin complexes (Supplementary Figure 8) and that excess DARPin suppresses this complex formation. Altogether, our structural and biochemical data provide a detailed molecular explanation on how kinesin-13 motors catalytically depolymerize MTs.

## Discussion
The structural and biochemical studies presented here fill several major gaps in our understanding of the MT depolymerization

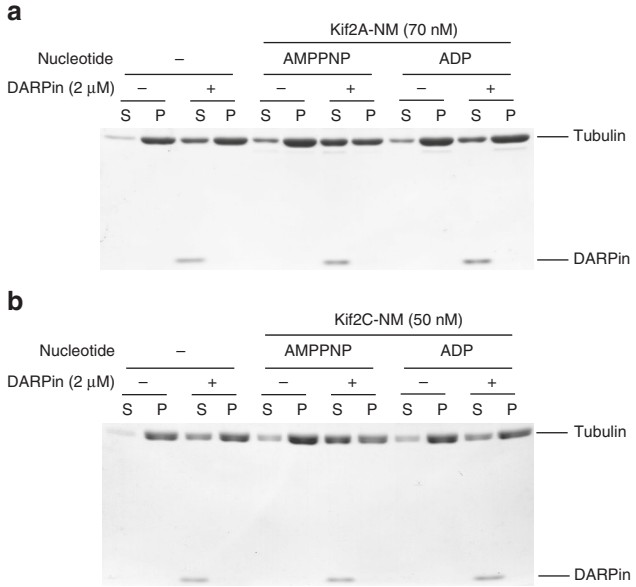

**Fig. 7** Nucleotide-dependent effect of DARPin on MT depolymerization by kinesin-13. **a**, **b** Depolymerization of taxol-stabilized MTs by Kif2A-NM (**a**) or Kif2C-NM (**b**) in the absence or presence of 2 μM DARPin and the indicated nucleotides (no nucleotide, AMPPNP, or ADP) was assessed by MT sedimentation assay. Samples were processed as described in Fig. 6f. A representative gel from at least three independent experiments is shown

mechanism of kinesin-13 motors. Our X-ray crystal structure of the Kif2A-NM–tubulin–DARPin complex reveals that ATP-bound Kif2A-NM binds two tubulin dimers and induces drastic bending of tubulin both intra-molecularly (within dimers) and inter-molecularly (between dimers). Here, the degree of tubulin curvature exceeds that of any tubulin oligomer reported to date[11,32,33,40,43,44]. Our biochemical and structural data provide evidence that this complex represents a trapped depolymerization intermediate of Kif2A-mediated MT depolymerization that is stable in solution prior to ATP hydrolysis. Importantly, this structure is distinct from the kinesin-13–tubulin complex structures reported in the two recently published studies[25,34] in two critical ways. First, it unambiguously shows the placement of the Kif2A neck against the second tubulin dimer. This interaction plays an indispensable role in forming the 1:2 Kif2A–tubulin complex by providing a critical anchor point for binding to the second tubulin dimer. Second, the crystal structure provides ample information to explain how Kif2A induces tubulin bending, especially at the interface between tubulin dimers. Guided and reinforced by neck and motor core interactions with the tubulin dimers on either side, the unique KVD motif of kinesin-13 inserts into the hydrophobic cavity at the inter-dimer interface and appears to displace the T7 loop and H8 helix of the $\alpha_2$-tubulin subunit. This conformational change may generate sufficient strain to weaken longitudinal interfaces within the MT lattice, leading to tubulin dissociation.

The Kif2A-NM–tubulin–DARPin structure also indicates that the structural change of the α4 helix of kinesin-13 motors upon tubulin binding is unremarkable and unlikely to mediate a direct effect for MT depolymerization. Moreover, our biochemical analysis of the Kif2A-MD construct clearly shows that the α4 helix and the class-specific KVD motif alone are insufficient to form the 1:2 complex and trigger MT depolymerization. In contrast, the kinein-13 neck peptide has been reported to induce lateral disintegration of MTs[45]. However, our Kif2A structure clearly shows that the neck forms a helical rather than a β-structure as reported[45]. It is likely that the neck plays a more

dominant role than the motor domain core to trigger drastic bending of two adjacent tubulin dimers, intra-molecularly and inter-molecularly. As MTs assembled in vitro have straight or slightly tapered ends[46–48], kinesin-13 motors likely first encounter a rather straighter protofilament, compared to those observed at depolymerizing ends[24,49]. This assessment is supported by the robust MT-stimulated ATPase activity of both kinesin-13-MD and -NM constructs. Upon binding to a MT end, neck binding may coordinate with the KVD motif to trigger the structural change needed for bending the adjacent tubulin dimers. While it is surprising that salt bridges are not observed between the charged residues of the KVD motif (Lys262 and Asp264) and the surface of tubulin in the Kif2A-NM–tubulin–DARPin complex, it is possible that they form upon initial binding of Kif2A to protofilaments at MT ends. These interactions could help guide the tip of loop 2 into the inter-dimer interface, and direct Val263 toward the full depth of its hydrophobic binding pocket on tubulin. At this point, additional bending of the tubulin protofilament would bring $\alpha_2$- and $\beta_1$-tubulin too close to the sides of the KVD finger to maintain the initial salt bridges with the tubulin surface. It is thus only through the combined action of the neck, KVD motif, and the motor domain core that kinesin-13 is able to induce extreme outward curvature of protofilament ends and trigger tubulin disassembly.

Combined with the rotational displacement of tubulin subunits, the extreme bending of adjacent tubulin dimers by Kif2A-NM may be sufficient to trigger dissociation of tubulin oligomers from the polymers because their curvature exceeds those typically observed at MT ends or within end-mimicking protofilament rings[50]. The fact that these drastically bent tubulins have not been directly observed by cryo-EM or in other crystallized structures suggests that they can no longer associate with the polymer or adjacent tubulins, and will rapidly revert back to the unconstrained curvature. In the Kif2A-NM–tubulin–DARPin structure presented here, this reversion is prevented by the presence of AMPPNP-trapped kinesin-13. Consistent with this assessment, our biochemical data show that Kif2A-NM is capable of depolymerizing MTs in the presence of AMPPNP and DARPin. On the other hand, the catalytic depolymerization (involving multiple cycles) requires ATP hydrolysis to release the kinesin-13 enzyme from the dissociated tubulin oligomers, as hinted by other studies[15,34]. Motivated by our structural information and SEC data with DARPin, we showed that this requirement can be bypassed by the use of DARPin in our in vitro MT depolymerization assay. These data provide further evidence that ATP binding to kinesin-13 rather than its hydrolysis is needed to trigger depolymerization at MT ends, likely by inducing more outward bending and destabilization of the curved protofilaments.

During the revision of this manuscript, Benoit et al.[51] reported high-resolution cryo-EM structures of *Drosophila* kinesin-13 KLP10A bound to curved or straight tubulin in different nucleotide states. These complexes show how conformational changes of the nucleotide pocket are coupled with movement of loop 2 in accord with the straightness of the tubulin polymer. Analogous to AMPPNP-Kif2A-NM, the nucleotide pocket of AMPPNP-KLP10A closes when in contact with curved tubulin, but remains open when bound to straight tubulin. Another similarity is that two-dimensional class average images of KLP10A-NM tubulin complexes show an elongated bar of density that emerges from the motor domain and extends to the next tubulin intra-dimer interface in virtually the same manner as helix α0a of the neck of Kif2A-NM. Although the authors suggest that the neck is not a major contributor to tubulin bending, our data on Kif2A-NM show otherwise. First, Kif2A-MD without the neck can neither form a 1:2 complex with tubulin nor depolymerize MT. Second, the degree of $\alpha_1\beta_1$-tubulin curvature induced

by Kif2A-NM exceeds those observed for KLP10A-NM. Together, the structural and biochemical data presented here help further define the molecular and mechanistic interplay between kinesin-13 and tubulin at MT ends.

## Methods

**Protein expression and purification.** Tubulin was isolated from bovine brain and purified by two cycles of polymerization-depolymerization in a high-molarity PIPES buffer as described by Castoldi et al.[52]. Purified tubulin was flash frozen in liquid $N_2$ and stored at $-80$ °C until use.

A plasmid coding Designed Ankyrin Repeat Protein, DARPin D1, was a kind gift from Dr. Jawdat Al-Bassam (University of California Davis). The gene was sub-cloned into the pET16b vector with NcoI and XhoI restriction enzymes, providing an N-terminal 6xHis-tag to DARPin. DARPin protein was expressed in Rosetta® GAMI B (DE3) Escherichia coli (Stratagene) cells in Luria-Bertani (LB) medium after induction with 0.5 mM isopropyl β-D-1-thiogalactopyranoside (IPTG) for 4 h at 37 °C. Cells were harvested by centrifugation, lysed by sonication, and DARPin was purified by Ni-NTA chromatography. Eluted protein was subjected to SEC on a Superdex 200 26/60 column (GE Healthcare) equilibrated with HEPES buffer (20 mM HEPES, 150 mM NaCl, and 1 mM $MgCl_2$, pH 7.2). Protein concentration was estimated by ultraviolet absorption using an extinction coefficient of 6990 $M^{-1}$ $cm^{-1}$. DARPin was concentrated in an Amicon® Ultra filter (Millipore, cutoff 10 kDa) to 20 mg $mL^{-1}$, flash frozen in liquid $N_2$, and stored at $-80$ °C.

The human Kif2A (isoform 3, UniProtKB—O00139) construct containing the neck and the motor domain (Kif2A-NM; residues 153–553) was originally generated by the Structural Genomics Consortium in Toronto and was obtained from Dr. Hernando Sosa (Albert Einstein Medical College, NY). Kif2A-NM protein expression was induced in BL21(DE3) pLysS E. coli (Promega) cells cultured in LB medium with 0.5 mM IPTG at 30 °C overnight. Cells were lysed in buffer A (50 mM Potassium phosphate, pH 8.0, 250 mM NaCl, 1 mM $MgCl_2$, 0.1% Triton X-100, 10 mM β-mercaptoethanol, and 10 mM imidazole) supplemented with 0.5 mM ATP and protease inhibitors. The lysate was then centrifuged, and the supernatant fraction was incubated with pre-equilibrated Ni-NTA resin (Qiagen) for 1–1.5 h. Histidine-tagged Kif2A-NM bound resin was then washed with buffer A with 0.5 M NaCl and 0.1 mM MgATP before eluting with buffer E (50 mM potassium phosphate, pH 7.0, 100 mM NaCl, 250 mM imidazole, 1 mM $MgCl_2$, 0.1 mM ATP, and 5 mM β-mercaptoethanol). Eluted fractions containing Kif2A-NM protein were pooled, diluted twofold in 20 mM HEPES buffer, pH 6.8, and absorbed onto a SP Sepharose Fast Flow column (GE Healthcare). The column was washed with buffer C (20 mM HEPES, pH 6.8, 50 mM NaCl, 1 mM dithiothreitol (DTT), 1 mM $MgCl_2$, and 0.2 mM ATP). After a quick final wash with buffer C without ATP, Kif2A-NM was eluted with buffer C with 0.5 M NaCl in the absence of ATP. The purified protein fractions were pooled and supplemented with at least molar equivalence of the desired nucleotide (e.g., AMPPNP) and concentrated to 50 mg $mL^{-1}$ using a Amicon® Ultra-4 filter (Millipore) in the final storage buffer (20 mM HEPES, pH 6.8, 0.5 M NaCl, 5 mM $MgCl_2$, 5 mM DTT, and ~2× molar equivalence of the desired nucleotide) and then flash frozen in liquid $N_2$.

Human Kif2A motor domain without the neck (denoted Kif2A-MD, residues 203–554) was amplified by PCR with the following primers (5′-GACTTAAGCTT GAATTCGACTTTAGAGGAAGTTTGGATTAT-3′ and 5′-CTGATATCGCGGC CGCTTAAGTCAATTCTTTGACCCTATTTG-3′), and cloned into a pGex-6P1 vector at the EcoRI and NotI sites. Protein expression and purification were performed essentially the same way as previously described for the KIF14MD_D772 construct[53]. The PreScission Protease-cleaved Kif2A-MD protein was used for this study. Human Kif2C-NM (187-589) and Kif2C-MD (255-589) were cloned, expressed, and prepared as previously described[54].

**Kif2A-NM–tubulin–DARPin complex formation.** Rapidly thawed tubulin solution (300 μL, 10 mg $mL^{-1}$) was mixed with 254.8 μL of HEPES buffer (20 mM HEPES, 150 mM NaCl, and 1 mM $MgCl_2$, pH 7.2) supplemented with 2.5 μL of 50 mM GDP and 42.7 μL of 20 mg $mL^{-1}$ DARPin, and incubated on ice for 10 min. Kif2A-NM (12.8 μL, 50 mg $mL^{-1}$) protein was diluted in 487 μL of HEPES buffer, and then slowly added to tubulin–DARPin solution in five small aliquots. The final Kif2A-NM–tubulin–DARPin mixture was then supplemented with AMPPNP to a final concentration of 0.1 mM and incubated on ice for 30 min. The molar ratio of Kif2A-NM:tubulin:DARPin was 0.8:1:1.05. This mixture was spun down for 10 min at 14 000 × g at 4 °C and 1 mL of supernatant was loaded onto a HiLoad 26/60 Superdex 200 column (GE Healthcare) equilibrated with HEPES buffer. AMPPNP (final conc. = 0.05 mM) was added to protein-containing fractions immediately after eluting from the column. The fractions containing the small (150 kDa) complex were concentrated down to an $A_{280}$ of ~10 using an Amicon® Ultra filter (Millipore, cutoff 50 kDa), flash frozen in liquid $N_2$, and stored at $-80$ °C.

**Analytical SEC.** Kif2A (2.57 nmol), tubulin (5.15 nmol), and DARPin (5.4 nmol) were mixed with GDP (final conc. 0.1 mM), and either AMPPNP, ATP, or ADP (0.1 mM) in a 150 μL total volume of HEPES buffer (HEPES 20 mM, NaCl 150 mM, and $MgCl_2$ 1 mM, pH 7.2). The mixture was incubated for 30 min on ice and then 100 μL of the sample was injected onto the column. SEC was performed on a

Superdex 200 10/300 GL chromatography column (GE Healthcare) equilibrated with HEPES buffer. The column was calibrated with molecular weight standards (GE Healthcare). Control experiments were performed with each protein alone. The collected fractions (500 μL) were concentrated by Amicon® Ultra filters (cutoff 10 kDa), then mixed with Laemmli buffer and resolved by SDS-PAGE.

**MT depolymerization assay.** MT preparation and sedimentation-based MT depolymerization assays were done essentially as described previously[54]. Briefly, Kif2A-NM or Kif2A-MD at the indicated concentrations were mixed with taxol-stabilized MT in BRB80-based depolymerization buffer (80 mM PIPES, pH 6.8, 1 mM $MgCl_2$, 1 mM EGTA, 20 μM taxol, 75 mM KCl, 0.25 mg $mL^{-1}$ bovine serum albumin, 1 mM DTT, and 0.02% Tween), supplemented with 1 mM ATP or the indicated nucleotide, and when indicated, with DARPin at the specified concentrations. Reactions were incubated at room temperature for 10 min unless specified otherwise. Free tubulin in solution was separated from remaining MTs pellet by centrifugation at 240,000 × g for 5 min at 25 °C. The supernatant fraction was retrieved from the sedimentation mixture and added to ¼ volume of 4× SDS loading buffer (Laemmli buffer). The remaining pellet was resuspended in an equal volume of 1× SDS loading buffer containing a similar buffer composition as the depolymerization buffer. Equal portions of the supernatant and pellet samples were resolved on SDS-PAGE. The gel was stained with Coomassie blue.

**ATPase activity assay.** A malachite green-based phosphate detection assay was used to measure kinesin-13-mediated ATPase activity, as previously described[54]. Briefly, reactions were assembled in the same buffer condition as in the depolymerization assay, with the indicated concentrations of tubulin/microtubules, kinesin-13 protein constructs, DARPin, and nucleotides. Reactions were allowed to proceed for the indicated length of time (usually 10–15 min, within the linear portion of the reaction curve), quenched with perchloric acid and malachite green reagent. The signal was quantified by the absorbance at 620 nm in a Genios Plus plate reader (Tecan).

**Crystallization and X-ray structure determination.** Crystals of the Kif2A-NM–tubulin–DARPin complex that were suitable for X-ray diffraction data collection grew in 3 days from 10-μL hanging drops containing the concentrated Kif2A-NM–tubulin–DARPin complex in a 1:1 ratio with a precipitant solution containing 8% PEG 8000, 6% ethylene glycol, 10 mM DTT, and 100 mM HEPES, pH 7.5 at 277 K. Prior to diffraction data collection, crystals were transferred into a cryoprotectant composed of 15% PEG 8000, 22% EG, and 100 mM HEPES, pH 7.5, and were then frozen in liquid $N_2$.

Diffraction data were collected from a single crystal at beamline 08ID-1 of the Canadian Light Source (Saskatoon, Canada) at 100 K, and were indexed, integrated, and scaled with HKL2000[55]. The structure was solved by MR with MolRep[56] using the structure of the tubulin subunits from the kinesin-1–tubulin complex (PDB ID: 4LNU) as a starting model. Once protein chains for one of the tubulin dimers were placed, interpretable density was visible for modeling a second set of tubulin chains and the DARPin molecule from the same structure (PDB ID: 4LNU). Next, the Kif2A molecule was found using PHASER[57] with provision of the tandem tubulin–tubulin complex obtained from MolRep, and the Kif2A-ADP structure (PDB ID: 2GRY) as separate ensembles. The Kif1A-AMPPNP structure (PDB ID: 1VFV) and kinesin-1–tubulin structure (PDB ID: 4LNU) were used to place AMPPNP, GDP, GTP, and $Mg^{2+}$ ions into the complex. The structure was refined with Phenix.refine[58] and manually optimized using COOT[59] to produce a final model with satisfactory $R_{work}/R_{free}$. The model quality was evaluated with MOLPROBITY[60], and by the wwPDB Validation Service. Data processing and refinement statistics are summarized in Table 1.

**SEC-SAXS data collection and analysis.** In-line SEC-SAXS measurements on the Kif2A-NM–tubulin–DARPin mixture were performed at the G1 Station of the Cornell High Energy Synchrotron Source[61] using 1.267 Å X-rays with a flux of $7.76 \times 10^{11}$ photons per second at a beam size of 250 μm x 250 μm. For the SEC-SAXS analysis, a mixture of Kif2A, tubulin, and DARPin was passed continuously through an X-ray sample cell via an in-line size-exclusion column (Superdex 200 10/300 GL, GE Healthcare) at a flow rate of 1 mL $min^{-1}$. The column was pre-equilibrated with running buffer consisting of 20 mM HEPES, 150 mM NaCl, and 1 mM $MgCl_2$, pH 7.2. The protein sample was prepared at 25 μM tubulin, 27 μM DARPin, 20 μM Kif2A-NM, and 0.1 mM AMPPNP and injected into a 100-μL loop.

Approximately 1200 two-second exposures were collected per sample, and 100 buffer profiles preceding the elution peaks were averaged and used for background subtraction. SAXS images were collected on dual Pilatus 100K-S detector system at sample-to-detector distances of 1.47 m. Samples were oscillated in the flow cell at 22 °C during data collection. SAXS data were processed using the BioXTAS RAW software[62]. The radius of gyration (Rg) and scattering intensity ($I(0)$) were calculated from the Guinier approximation, and the pair distribution function, P(r), was calculated by GNOM[63]. The dummy atom model of the Kif2A-NM–tubulin–DARPin complex was calculated using DAMMIF[64]. Ten independent dummy atom models were averaged and selected using DAMAVER[65]. The resulting experimental SAXS profile was then compared with simulated

(theoretical) scattering curve of the Kif2A-NM–tubulin–DARPin complex crystal structure using the program CRYSOL[66].

**Data availability**. Coordinates and structure factors have been deposited in the Protein Data Bank with accession code: 6BBN. SAXC data and models have been deposited in the Small-Angle Scattering Biological Data Bank with accession code: SASDCR9. Other data are available from the corresponding authors upon reasonable request.

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

## Acknowledgements

We thank Peter L. Davies, Rachel Trister, Jacqueline Hellinga, Byron Hunter, Tyler Vance, Irsa Shoukat, and Robert Campbell for technical assistance and help with preparation of the manuscript. We thank Dr. Jawdat Al-Bassam, (University of California Davis) for providing a plasmid with the DARPin D1 gene. Diffraction data were collected at the Canadian Light Source, Saskatoon, Canada (beamline 08ID-1), and we thank the beamline group for making these experiments possible. This work was supported by funding to J.S.A. and B.H.K. from CCSRI, CIHR and NSERC. J.S.A. holds a Canada Research Chair (Tier 2) in Structural Biology. B.H.K. is a recipient of the Fonds de recherche du Québec—Santé (FRQS) Chercheure-boursière Junior 1 and Junior 2 Awards and the Canadian Institutes of Health Research (CIHR) New Investigator Award. The Institute for Research in Immunology and Cancer (IRIC) is supported in part by the Canadian Center of Excellence in Commercialization and Research (CECR), the Canada Foundation for Innovation and FRQS.

## Author contributions

D.T., B.H.K. and J.S.A. designed research; D.T., M.P., L.T. and B.H.K. purified protein reagents, characterized the kinesin–tubulin interaction biochemically; D.T. and J.S.A. crystallized the complex and determined its structure; D.T., M.P., A.Z., B.H.K. and J.S.A. analyzed the data; D.T., B.H.K. and J.S.A. wrote the manuscript with inputs from all authors.

## Additional information

**Competing interests:** The authors declare no competing interests.

