## [Peer Review File · Nature Communications]

Reviewers' comments:

Reviewer #1 (Remarks to the Author):

The authors have identified, separated out and crystallised a large complex that consists of a kinesin-13 motor protein bound to TWO tubulin heterodimers. This means they can see how the motor bonds to a curved protofilament. The work is an important new step in understanding how the motor protein stabilises the curved protofilament conformation and thereby promotes microtubule disassembly. The measurements directed at understanding the reason for ATPase activity are also interesting. Presumably it occurred first in depolymerising kinesin to maximise the activity of a low number of molecules and was subsequently adopted for movement along a MT

Reviewer #2 (Remarks to the Author):

The manuscript of Trofimova et al describes work to investigate the molecular mechanism of kinesin-13 catalyzed microtubule depolymerization. While substantial progress has been made in elucidating the molecular mechanisms of several other members of the kinesin superfamily of molecular motors, the way(s) in which kinesin-13s promote microtubule catastrophe are poorly understood by comparison. Trofimova et al present a crystal structure of a depolymerization-active construct of the kinesin-13 Kif2A – the so-called Kif2A-neck+motor construct (Kif2A-NM) - with AMPPNP bound, in complex with two longitudinally connected tubulin dimers, and stabilized by a tubulin-specific DARPin. While structures of kinesin-13 motor domains in complex with tubulin dimers have been previously published, the authors' structure is remarkable in the visualization of the neck region along the surface of the more minus-end located tubulin dimer and the conformation of this tubulin dimer. The structure supports the previously described stoichiometry of NM-containing constructs bound to curved 2x tubulin complexes and sheds light on the contribution of this region of the motor in microtubule depolymerization. The authors provide some biochemical data to support the mechanistic relevance of their structure, which is largely well-aligned with prior knowledge concerning kinesin-13 structure and mechanism. This includes further insight into the role of ATP binding compared to ATP hydrolysis in kinesin-13 mediated depolymerization. While the captured conformation of the motor-tubulin complex is intriguing, the manuscript overall is incomplete in the treatment of the mechanistic questions raised by the structure.

- 1) The introduction text states that the previously solved structures of kinesin-13 motor domains are very similar to those of other non-depolymerizing kinesins, but no analysis is provided to support this statement. It is also not consistent with others' accounts - e.g. Ogawa et al, 2004; 2017 - linking a more curved tubulin interface of kinesin-13s to their microtubule depolymerization capabilities. This should be explained or refuted more fully;
- 2) The result section begins with a convoluted account of the experimental path to purification and characterization of the crystallized complex – the text here is not very clear and could be shortened to allow focus to be brought to the structure itself;
- 3) The conformation of the tubulin dimer to which the kinesin-13 motor domain itself is bound is very similar to that in other tubulin crystal structures. The authors conclude that the kinesin-13 motor domain does not induce tubulin dimer bending, but this ignores the context in which kinesin-13 depolymerization occurs, which is at microtubule ends where the terminal tubulins may not be as curved as tubulin in solution (see numerous studies by Chretien et al). This aspect of the kinesin-13 activity needs to be described more precisely;
- 4) There is almost no account provided of the conformation of the kinesin-13 motor domain in the crystal structure apart from the disembodied and confusing description of the conformation of helix-alpha4. A more complete account should be included, however briefly, including a characterization of the active site;
- 5) The authors also ignore the EM data from the Milligan and Sosa labs in which kinesin-13 motor domain constructs lacking the neck region were shown to depolymerize microtubules (Moores et

al, 2002; Tan et al, 2008). How can these results be reconciled with the authors' neck-focused mechanism?

6) The tubulin dimer to which the neck region is bound adopts a more extreme curvature than has been previously described, but how this curvature is accommodated within the structure of the tubulin dimer is not described.

7) The authors also describe rotational curvature of the tubulin dimers in their structure (Fig3, bottom). How does that compare with that seen in other tubulin crystal structure (e.g. Structural plasticity of tubulin assembly probed by vinca-domain ligands. Ranaivoson FM, Gigant B, Berritt S, Joullié M, Knossow M. *Acta Crystallogr D Biol Crystallogr*. 2012 Aug; 68(Pt 8):927-34.

8) The authors should refer to the neglected work of Shimizu et al in their interpretations: Effects of the KIF2C neck peptide on microtubules: lateral disintegration of microtubules and β -structure formation. Shimizu Y, Shimizu T, Nara M, Kikumoto M, Kojima H, Morii H. *FEBS J*. 2013 Apr; 280(7):1681-92.

9) The authors refer to other conformations of kinesin-13 neck and loop2 regions that have been previously structurally characterized but it is not clear from their account how these different conformations fit together into a coherent mechanistic picture;

Minor points

- The introduction provides a relatively detailed description of previous work in the field and could be usefully summarized for the benefit of the more general Nature Communications reader;
- In Fig3, a secondary structural element in each of the curved alpha- and beta-tubulin subunits are depicted in a dark color – what is it and why is it colored in this way?
- The figures need a greater consistency of depiction for ease of understanding – e.g. in Fig4 the effective microtubule plus end is both towards the top (c, d, left) and bottom (d, right) of the page;

Reviewer #3 (Remarks to the Author):

This is an important paper that describes a key missing piece of the structural interactions that define the functioning of the kinesin-13 family of microtubule depolymerases. The authors do this by crystallizing a functional complex with a fragmented microtubule protofilament (two adjacent tubulin dimers). A growing body of structural work has studied this motor family, including the very recent co-complex of a truncated form of KIF2C that lacked the full neck. However, until now, the role of the "neck" region of these motors, which is required for depolymerase activity, has only been visualized in a presumed non-functional orientation with respect to tubulin (in a single structure, from Gigant's group- Ref. 35 of this manuscript). This new work combines a groundbreaking new structure - KIF2B including a complete neck, co-complexed with two dimers of tubulin - with an innovative biochemical study that illuminates the role of ATP hydrolysis in the reaction. The authors creatively took advantage of the same microtubule depolymerizing factor (DARPin) that was used to produce crystals, to strongly indicate that hydrolysis is not required for the key step in which tubulin is "peeled" off the microtubule. By a clever series of experiments they came up with an elegant mechanism in which the DARPin essentially wedges apart the two tubulin dimers to which a KIF2B binds (using the motor domain and neck, respectively), in order to dissociate the KIF2B from tubulin even in the presence of AMPPNP where the hydrolysis cycle cannot reach the final, weak-binding ADP state. These findings have strong relevance to the problem of how the kinesin-family microtubule polymerases work structurally, and will have broad interest given the relative scarcity of structural information on the actions of cytoskeletal depolymerizing factor generally.

I think the manuscript is relatively well-polished and do not find any major problems with it. I think it is essentially ready to accept as is, but I do offer a few minor suggestions.

1. I am not so sure about the authors' proposal that insertion of the "V" from "KVD" into the a pocket in alpha1 tubulin directly displaces T7/H7 region, destabilizing the tubulin inter-dimer

interface and facilitating depolymerization. If so, this should have been seen in the KIF2C/MCAK/DARPin complex with a single tubulin dimer (Ref. 35 from Gigant's group), but it is stated on the top of p. 14 that this is not the case. Moreover, if KVD by itself could perturb T7/H8, wouldn't a neck-less Kin13 have some depolymerization activity? I think it is equally likely that the T7/H8 distortion results from the tubulin bending (caused, as the authors propose, by neck binding). I recommend the authors consider this other hypothesis, which does not diminish their findings.

2. On P13 it is stated:

"of the KVD motif [...] H-bonds or salt bridges are not observed in the Kif2A-tubulin-DARPin complex in the manner that was predicted in modeling studies of Kif2C22"

However, it appears from the figures that these interactions in the reported structure agree very well with the DARPin-Kif2C tubulin cocrystal from Gigant's group (5MIO); this should be stated.

3. I found the section: "Relationship between ATP turnover by KIF2A and MT depolymerization' (P. 15) hard to follow. In particular:

(A) There is a series of observations regarding regimes of linearity:

(P 15) "We chose ... linear turnover rate ($> 1\mu\text{M}$ tubulin dimer)"... resulting in MT concentration below $1\mu\text{M}$."

"DARPIN may act synergistically to accelerate ... MT depolymerization... resulting in MT concentration below $1\mu\text{M}$ "

(P 15) "the ATPase rates of our control experiment without DARPin stayed relatively linear over the 10-min time period"

But it was quite difficult for me to piece this logic together. In the end, I think I understood it but the reasoning doesn't seem to be laid out as clearly as it could be.

This section winds up with what I think is an important observation that DARPin could compete KIF2B off the double tubulin dimer:

(P 17) "...DARPin can bypass this requirement [for hydrolysis in order to release KIF2A from tubulin dimers] by releasing the enzyme from the bound tubulin dimers via competition with its neck binding"

However, the described mechanism is hard to follow, since it doesn't look to me like the site of DARPin binding collides with the neck at all. Seems to me that what must be going on is that the DARPin wedges apart the two tubulin dimers- maybe disrupting the KVD site, but this would be indirect since the DARPin interface is actually with beta tubulin and not alpha tubulin.

(B) Right afterwards (P 17), it is said:

"Additional support for this can be derived from our observation that AMP-PNP, and not ATP or ADP, is able to produce stable 1:2:1 Kif2A-tubulin-DARPin complexes (Fig. S5)."

-> It would be helpful to spell out the logic here. At first glance, the observation of a stable 1:2:1 Kif2A:tubulin:DARPin complex seems to run counter to the proposal that DARPin *releases* AMPPNP-bound KIF2A from the tubulin dimer. Maybe I'm just a bit slow here but I had to think about this for a while before realizing that the authors are suggesting that an additional DARPin would infiltrate the 1:2:1 complex to dissociate the tubulin dimers and dislodge the neck.

Minor corrections/grammar:

While this is more stylistic and doesn't interfere so much with communication of the message, there is an overuse of the pronoun "this" without an accompanying, descriptive noun. For example:

Abstract

This signifies that the crystallized Kif2A-tubulin complex
"This" -> "These results"

Introduction

"This is essential" -> "This activity is essential"

P4: "This is supported by" -> "This idea is supported by"

P10: "This demonstrates that depolymerization-competent"

P11: "This is interesting because [...]" -> "In contrast, [...] of the tubulin dimer in reported co-complexes."

P12: "This explains why"

"This suggests that"

P13: "While this appears to be inconsistent with"

P17: "Additional support for this"

Responses to Reviewers' Comments

We thank all the reviewers for their extensive reviews and valuable suggestions, which are very much appreciated. We have revised our manuscript accordingly to address all the points raised by the reviewers. Please also refer to our point-to-point responses below.

Reviewers' comments:

Reviewer #1 (Remarks to the Author):

The authors have identified, separated out and crystallised a large complex that consists of a kinesin-13 motor protein bound to TWO tubulin heterodimers. This means they can see how the motor bonds to a curved protofilament. The work is an important new step in understanding how the motor protein stabilises the curved protofilament conformation and thereby promotes microtubule disassembly. The measurements directed at understanding the reason for ATPase activity are also interesting. Presumably it occurred first in depolymerising kinesin to maximise the activity of a low number of molecules and was subsequently adopted for movement along a MT

Response:

We are pleased to see the positive remarks made by Reviewer #1 and find his/her hypothesis interesting.

Reviewer #2 (Remarks to the Author):

The manuscript of Trofimova et al describes work to investigate the molecular mechanism of kinesin-13 catalyzed microtubule depolymerization. While substantial progress has been made in elucidating the molecular mechanisms of several other members of the kinesin superfamily of molecular motors, the way(s) in which kinesin-13s promote microtubule catastrophe are poorly understood by comparison. Trofimova et al present a crystal structure of a depolymerization-active construct of the kinesin-13 Kif2A – the so-called Kif2A-neck+motor construct (Kif2A-NM) - with AMPPNP bound, in complex with two longitudinally connected tubulin dimers, and stabilized by a tubulin-specific DARPin. While structures of kinesin-13 motor domains in complex with tubulin dimers have been previously published, the authors' structure is remarkable in the visualization of the neck region along the surface of the more minus-end located tubulin dimer and the conformation of this tubulin dimer. The structure supports the previously described stoichiometry of NM-containing constructs bound to curved 2xtubulin complexes and sheds light on the contribution of this region of the motor in microtubule depolymerization. The authors provide some biochemical data to support the mechanistic relevance of their structure, which is largely well-aligned with prior knowledge concerning kinesin-13 structure and mechanism. This includes further insight into the role of ATP binding compared to ATP hydrolysis in kinesin-13 mediated depolymerization. While the captured conformation of the motor-tubulin complex is intriguing, the manuscript overall is incomplete in the treatment of the mechanistic

questions raised by the structure.

We thank this reviewer for recognizing our work and making extensive and helpful suggestions. We hope he/she will find our work more complete in the revised version. Please see our point-by-point responses below.

1) The introduction text states that the previously solved structures of kinesin-13 motor domains are very similar to those of other non-depolymerizing kinesins, but no analysis is provided to support this statement. It is also not consistent with others' accounts - e.g. Ogawa et al, 2004; 2017 - linking a more curved tubulin interface of kinesin-13s to their microtubule depolymerization capabilities. This should be explained or refuted more fully;

We have removed the sentence from the Introduction stating that structures of kinesin-13 motor domains are very similar to those of other non-depolymerizing kinesins in order to address this issue more fully in the Results section at the top of Page 12.

We would also like to note in this response that Ogawa *et al.* (2004) indicated in their manuscript that the majority of the Kif2C motor domain is nearly identical to Kif1A and other KIFs.

With regard to microtubule depolymerization capabilities being linked to a more curved tubulin interface on kinesin-13s, we are not convinced that the 2004 and 2017 studies by Ogawa et al. sufficiently support this idea for the following reasons:

In silico docking of the Kif2C-AMPPNP structure (not complexed with tubulin) onto a MT protofilament did not take into account the rotation of helices $\alpha 4$ and $\alpha 5$ that occur in both kinesin-13 and kinesin-1 upon tubulin binding (Ogawa *et al.* 2004). Indeed, their structure AMPPNP-bound Kif2C is essentially the same as ADP-Kif2C.

In their 2017 publication, Ogawa *et al.* submitted that they had identified the conformation in which the KIF2 core domain binds tightly to two tubulin dimers in the middle pre-hydrolysis state during ATP hydrolysis. This was determined by crystallizing mouse KIF2C-ADP-BeFx and KIF2C-ADP-AIFx monomers that were purified from 1:2 and 1:1 complexes with tubulin, but did not actually contain tubulin in the crystal. When we superimpose these structures against our tubulin-bound Kif2A-NM-AMPPNP structure, there are substantial differences in their tubulin-binding surface compared to Kif2A. To illustrate this, we have added a figure (Fig. S5) in which the structure of the KIF2C-ADP-BeFx monomer is superimposed via its P-loop onto Kif2A of the 1:2 Kif2A-tubulin structure, and have quantify their dissimilarity. Therefore, we suggest (on Page 12) that these structures are not informative of how the kinesin-13 core recognizes or stabilizes the curved conformation of two tubulin dimers.

2) The result section begins with a convoluted account of the experimental path to

purification and characterization of the crystallized complex – the text here is not very clear and could be shortened to allow focus to be brought to the structure itself;

We very much appreciated this suggestion and have revised our text accordingly (please see the corresponding section on pp. 6-7).

3) The conformation of the tubulin dimer to which the kinesin-13 motor domain itself is bound is very similar to that in other tubulin crystal structures. The authors conclude that the kinesin-13 motor domain does not induce tubulin dimer bending, but this ignores the context in which kinesin-13 depolymerization occurs, which is at microtubule ends where the terminal tubulins may not be as curved as tubulin in solution (see numerous studies by Chretien et al). This aspect of the kinesin-13 activity needs to be described more precisely;

In vitro, MT ends are indeed not as bent as tubulin dimers in solution. Although it is tempting to speculate kinesin-13s recognize the curvature at MT ends, our view is that they bind to both straight (MT lattice) and curved protofilaments (in protofilament ends). We argue that when kinesin-13 binds to tubulin dimers, it induces curvature that exceeds those observed at MT ends and those of tubulin dimers in solution, such that they can no longer associate with MT ends. This point is now more explicitly and extensively discussed in the revised text (p. 18).

4) There is almost no account provided of the conformation of the kinesin-13 motor domain in the crystal structure apart from the disembodied and confusing description of the conformation of helix-alpha4. A more complete account should be included, however briefly, including a characterization of the active site;

We have created a separate section in the Results that describes the overall conformation of the Kif2A motor domain and the active site (please see pp. 11-12). We have also added a new Figure depicting the motor domain conformation in comparison to the isolated ADP-Kif2A structure (Figure 4).

5) The authors also ignore the EM data from the Milligan and Sosa labs in which kinesin-13 motor domain constructs lacking the neck region were shown to depolymerize microtubules (Moores et al, 2002; Tan et al, 2008). How can these results be reconciled with the authors' neck-focused mechanism?

The reason we did not contrast our work with the Moores *et al*, 2002 paper is because the fungal kinesin-13/pKinI from *P. falciparum* differs from vertebrate kinesin-13 in its requirement of the neck to depolymerize MTs (this point has been added to the introduction (p.5). Therefore, we did not make any direct comparison with it for the rest of the manuscript. In the case of the Tan *et al*, 2008 paper, both of the human kinesin-13 constructs that they used do indeed contain either the full neck (HsKif2A) or portion of the neck (HsKif2C) that is sufficient to depolymerize MTs. In fact, we are using the same HsKif2A construct (i.e. Kif2A-NM) for our study. Therefore, our results do not contradict theirs, although we disagree with their interpretations.

6) The tubulin dimer to which the neck region is bound adopts a more extreme curvature than has been previously described, but how this curvature is accommodated within the structure of the tubulin dimer is not described.

After comparing the conformation of each tubulin subunit of the Kif2A-tubulin-DARPin complex with many the available tandem tubulin complex crystal structures, we identified a grouping of structural differences at each tubulin interface that could explain how additional curvature and rotation are accommodated. The second paragraph on p. 13 has been devoted to describing these structural differences, using the stathmin-tubulin-cholchicine complex (PDB ID: 1SA0) as a reference model.

7) The authors also describe rotational curvature of the tubulin dimers in their structure (Fig3, bottom). How does that compare with that seen in other tubulin crystal structure (e.g. Structural plasticity of tubulin assembly probed by vinca-domain ligands. Ranaivoson FM, Gigant B, Berritt S, Joullié M, Knossow M. Acta Crystallogr D Biol Crystallogr. 2012 Aug;68(Pt 8):927-34.

We have included a new figure (Figure S6) comparing the curvature and rotation of tubulin subunits of the Kif2A complex to those of straight tubulin and two stathmin-tubulin complexes containing vinca-domain ligands. This comparison is described at the top of p.13.

8) The authors should refer to the neglected work of Shimizu et al in their interpretations: Effects of the KIF2C neck peptide on microtubules: lateral disintegration of microtubules and β -structure formation. Shimizu Y, Shimizu T, Nara M, Kikumoto M, Kojima H, Morii H. FEBS J. 2013 Apr;280(7):1681-92.

We thank for the reviewer for pointing this out. The reference is now added and discussed in the Discussion section in the middle of p.18.

9) The authors refer to other conformations of kinesin-13 neck and loop2 regions that have been previously structurally characterized but it is not clear from their account how these different conformations fit together into a coherent mechanistic picture;

This is an excellent point, and unfortunately we do not yet understand how these other conformations of the neck relate to the way in which kinesin-13s work because they do not interact with the surface of tubulin or the motor domain (other than via crystal contacts) and the longer of the two neck helices (α 0a) is not ordered. Nonetheless, we have now included an expanded description of these alternate conformations on pp. 9-10 of the Results.

Minor points

- The introduction provides a relatively detailed description of previous work in the field and could be usefully summarized for the benefit of the more general Nature Communications reader;

We have revised the Introduction section to remove some of the more extraneous details as suggested, keeping only those directly relevant to the current study.

- In Fig3, a secondary structural element in each of the curved alpha- and beta-tubulin subunits are depicted in a dark color – what is it and why is it colored in this way?

This was done to provide a point of reference with which to emphasize the bend and twist in the protofilament that is created by Kif2A. We have now indicated this in the figure legend (now Figure 5).

- The figures need a greater consistency of depiction for ease of understanding – e.g. in Fig4 the effective microtubule plus end is both towards the top (c, d, left) and bottom (d, right) of the page;

We have modified our figures so that they present the model in orientations that are more straightforward to interpret, and have added labels to the MT ends. Note that the former Fig. 4 is now Fig. 3.

Reviewer #3 (Remarks to the Author):

This is an important paper that describes a key missing piece of the structural interactions that define the functioning of the kinesin-13 family of microtubule depolymerases. The authors do this by crystallizing a functional complex with a fragmented microtubule protofilament (two adjacent tubulin dimers). A growing body of structural work has studied this motor family, including the very recent co-complex of a truncated form of KIF2C that lacked the full neck. However, until now, the role of the "neck" region of these motors, which is required for depolymerase activity, has only been visualized in a presumed non-functional orientation with respect to tubulin (in a single structure, from Gigant's group- Ref. 35 of this manuscript). This new work combines a groundbreaking new structure - KIF2B including a complete neck, co-complexed with two dimers of tubulin - with an innovative biochemical study that illuminates the role of ATP hydrolysis in the reaction. The authors creatively took advantage of the same microtubule depolymerizing factor (DARPin) that was used to produce crystals, to strongly indicate that hydrolysis is not required for the key step in which tubulin is "peeled" off the microtubule. By a clever series of experiments they came up with an elegant mechanism in which the DARPin essentially wedges apart the two tubulin dimers to which a KIF2B binds (using the motor domain and neck, respectively), in order to dissociate the KIF2B from tubulin even in the presence of AMPPNP where the hydrolysis cycle cannot reach the final, weak-binding ADP state. These findings have strong relevance to the problem of how the kinesin-family microtubule polymerases work structurally, and will have broad interest given the relative scarcity of structural information on the actions of cytoskeletal depolymerizing factor generally.

We thank the reviewer for recognizing the importance our work. This is very much appreciated.

I think the manuscript is relatively well-polished and do not find any major problems with it. I think it is essentially ready to accept as is, but I do offer a few minor suggestions.

1. I am not so sure about the authors' proposal that insertion of the "V" from "KVD" into a pocket in alpha1 tubulin directly displaces T7/H7 region, destabilizing the tubulin inter-dimer interface and facilitating depolymerization. If so, this should have been seen in the KIF2C/MCAK/DARPin complex with a single tubulin dimer (Ref. 35 from Gigant's group), but it is stated on the top of p. 14 that this is not the case. Moreover, if KVD by itself could perturb T7/H8, wouldn't a neck-less Kin13 have some depolymerization activity? I think it is equally likely that the T7/H8 distortion results from the tubulin bending (caused, as the authors propose, by neck binding). I recommend the authors consider this other hypothesis, which does not diminish their findings.

We were very excited by this finding and proposed that displacement of the T7-H8 motif was not visible in the Kif2C structure by Gigant *et al.* because that structure lacked the second tubulin dimer and the interaction of the neck with those subunits. To more accurately describe the potential significance of this finding, we have described the outcome of our comparison of the KifA-tubulin complex with other tandem tubulin structures on pp. 10-11. We have also modified the figure showing the T7-H8 conformational change (now Figure 3d). The effect of this change on the adjacent tubulin subunit is also described on p. 13.

2. On P13 it is stated:

"of the KVD motif [...] H-bonds or salt bridges are not observed in the Kif2A-tubulin-DARPin complex in the manner that was predicted in modeling studies of Kif2C22"

However, it appears from the figures that these interactions in the reported structure agree very well with the DARPin-Kif2C tubulin cocrystal from Gigant's group (5MIO); this should stated.

In that section of the Results (now in the middle of p.10), we were referring to the Wang *et al.* (2015) JBC paper involving computational modeling studies of Kif2C, not the recent Kif2C-tubulin complex structure. We have revised this description on p.10 to clarify this, and to acknowledge that the Kif2C-tubulin complex (PDB: 5MIO) does exhibit a similar arrangement of the L2 residues (KVD) to Kif2A-tubulin.

3. I found the section: "Relationship between ATP turnover by KIF2A and MT depolymerization' (P. 15) hard to follow. In particular:

We thank the reviewer for this candid comment. The text has now been revised to clarify the description (please see the revised text on pp.14-15 and below).

(A) There is a series of observations regarding regimes of linearity:

(P 15) "We chose ... linear turnover rate ($> 1\mu\text{M}$ tubulin dimer)" ... resulting in MT concentration below $1\mu\text{M}$."

"DARPin may act synergistically to accelerate ... MT depolymerization... resulting in MT concentration below $1\mu\text{M}$ "

(P 15) "the ATPase rates of our control experiment without DARPin stayed relatively linear over the 10-min time period"

But it was quite difficult for me to piece this logic together. In the end, I think I understood it but the reasoning doesn't seem to be laid out as clearly as it could be.

We agree with the reviewer's assessment. We have thus replaced the regimes of "linearity" with a more explicit and clearer description on p. 15. We hope the revised text clarifies our reasoning.

This section winds up with what I think is an important observation that DARPin could compete KIF2B off the double tubulin dimer:

(P 17) "...DARPin can bypass this requirement [for hydrolysis in order to release KIF2A from tubulin dimers] by releasing the enzyme from the bound tubulin dimers via competition with its neck binding". However, the described mechanism is hard to follow, since it doesn't look to me like the site of DARPin binding collides with the neck at all. Seems to me that what must be going on is that the DARPin wedges apart the two tubulin dimers- maybe disrupting the KVD site, but this would be indirect since the DARPin interface is actually with beta tubulin and not alpha tubulin.

We agree completely with the reviewer and think that their description provides a more accurate explanation for how excess DARPin may be affecting the complex. We have edited the revised manuscript accordingly on p.8 & pp. 16-17 of the Results section.

(B) Right afterwards (P 17), it is said:

"Additional support for this can be derived from our observation that AMP-PNP, and not ATP or ADP, is able to produce stable 1:2:1 Kif2A-tubulin-DARPin complexes (Fig. S5)."

-> It would be helpful to spell out the logic here. At first glance, the observation of a stable 1:2:1 Kif2A:tubulin:DARPin complex seems to run counter to the proposal that DARPin *releases* AMPPNP-bound KIF2A from the tubulin dimer. Maybe I'm just a bit slow here but I had to think about this for a while before realizing that the authors are

suggesting that an additional DARPin would infiltrate the 1:2:1 complex to dissociate the tubulin dimers and dislodge the neck.

This is correct, and we have revised our description of the probable mode of action of DARPin on pp. 16-17 to reflect this reviewer's assessment. Here, we have highlighted the concept that although DARPin does not bind to the same site as the Kif2A neck, its association with β_1 -tubulin is incompatible with simultaneous binding of the neck helix and thereby displacing the $\alpha_1\beta_1$ -tubulin dimer from the tandem ternary complex.

Minor corrections/grammar:

While this is more stylistic and doesn't interfere so much with communication of the message, there is an overuse of the pronoun "this" without an accompanying, descriptive noun. For example:

Abstract

This signifies that the crystallized Kif2A-tubulin complex
"This" -> "These results"

Corrected.

Introduction

"This is essential" -> "This activity is essential"

Corrected.

P4: "This is supported by" -> "This idea is supported by"

Corrected.

P10: "This demonstrates that depolymerization-competent"

Corrected.

P11: "This is interesting because [...]" -> "In contrast, [...] of the tubulin dimer in reported co-complexes."

Corrected.

P12: "This explains why"

"This suggests that"

Corrected.

P13: "While this appears to be inconsistent with"

P17: "Additional support for this"

Corrected.

REVIEWERS' COMMENTS:

Reviewer #2 (Remarks to the Author):

The revisions undertaken by Trofimova et al have significantly improved their manuscript and clarified the insights their data provide into the molecular mechanism of kinesin-13-catalyzed microtubule depolymerization. This work represents an important contribution to the literature. There have been several recent studies about kinesin-13 mechanism which Trofimova et al place in useful context of their own work. A study from the Sosa lab (Benoit et al, Nat Comms 2018) is so recent that it is currently not cited by Trofimova et al. However, the two studies provide some overlapping insights about aspects of the kinesin-13 mechanism, although in particular the data of Trofimova et al about the structure of the neck region and its effects on tubulin conformation are more detailed and informative. However it would be invaluable for Trofimova et al to add a paragraph in the Discussion comparing their data with this recent publication.

Minor point

Legend to Fig 4 should be edited because e.g. a) does not show a view from the tubulin surface, b) does not show a close up of the nucleotide-binding site

REVIEWERS' COMMENTS:

Reviewer #2 (Remarks to the Author):

The revisions undertaken by Trofimova et al have significantly improved their manuscript and clarified the insights their data provide into the molecular mechanism of kinesin-13-catalyzed microtubule depolymerization. This work represents an important contribution to the literature. There have been several recent studies about kinesin-13 mechanism which Trofimova et al place in useful context of their own work. A study from the Sosa lab (Benoit et al, Nat Comms 2018) is so recent that it is currently not cited by Trofimova et al. However, the two studies provide some overlapping insights about aspects of the kinesin-13 mechanism, although in particular the data of Trofimova et al about the structure of the neck region and its effects on tubulin conformation are more detailed and informative. However it would be invaluable for Trofimova et al to add a paragraph in the Discussion comparing their data with this recent publication.

Minor point

Legend to Fig 4 should be edited because e.g. a) does not show a view from the tubulin surface, b) does not show a close up of the nucleotide-binding site

Responses to Reviewers' Comments

We are very pleased by these positive remarks and have used the final suggested revisions to frame our study in the context of the most up-to-date literature on the mechanism of kinesin-13s. The end of our Discussion now includes a paragraph detailing the overlapping and distinct insights of our study and that of Benoit et al, Nat Comms 2018.

We have also corrected the description in the legend to Figure 4.